# Understanding Parametric and Contextual Knowledge Reconciliation within Large Language Models

**Jun Zhao**[1,4], **Yongzhuo Yang**[1], **Xiang Hu**[4], **Jingqi Tong**[1], **Yi Lu**[1],
**Wei Wu**[4*], **Tao Gui**[1,3], **Qi Zhang**[1,2*], **Xuanjing Huang**[1,2]
[1]College of Computer Science and Artificial Intelligence, Fudan University
[2]Shanghai Collaborative Innovation Center of Intelligent Visual Computing
[3]Shanghai Innovation Institute    [4]Ant Group
{zhaoj19,qz}@fudan.edu.cn, wuwei19850318@gmail.com

## Abstract

Retrieval-Augmented Generation (RAG) provides additional *contextual knowledge* to complement the *parametric knowledge* in Large Language Models (LLMs). These two knowledge interweave to enhance the accuracy and timeliness of LLM responses. However, the internal mechanisms by which LLMs utilize these knowledge remain unclear. We propose modeling the forward propagation of knowledge as an *entity flow*, employing this framework to trace LLMs' internal behaviors when processing mixed-source knowledge. Linear probing utilizes a trainable linear classifier to detect specific attributes in hidden layers. However, once trained, a probe cannot adapt to dynamically specified entities. To address this challenge, we construct an entity-aware probe, which introduces special tokens to mark probing targets and employs a small trainable rank-8 lora update to process these special markers. We first verify this approach through an attribution experiment, demonstrating that it can accurately detect information about ad-hoc entities from complex hidden states. Next, we trace entity flows across layers to understand how LLMs reconcile conflicting knowledge internally. Our probing results reveal that contextual and parametric knowledge are routed between tokens through distinct sets of attention heads, supporting attention competition only within knowledge types. While conflicting knowledge maintains a residual presence across layers, aligned knowledge from multiple sources gradually accumulates, with the magnitude of this accumulation directly determining its influence on final outputs.

## 1 Introduction

Through large-scale pre-training, Large Language Models (LLMs) have acquired vast amounts of knowledge implicitly encoded within their model parameters, forming the cornerstone of their general capabilities. However, this paradigm of knowledge storage faces twofold limitations: from a coverage perspective, *parametric knowledge* obtained through pre-training on public data struggles to extend to domain-specific private knowledge; from a temporal perspective, static parametric knowledge gradually loses its effectiveness as the real world evolves. To address these limitations, RAG techniques have been introduced to provide LLMs with additional contextual support [8, 45]. Such real-time *contextual knowledge* both expands the inherent boundaries of parametric knowledge and mitigates its temporal constraints.

This raises a crucial question: to what extent can LLMs accept and utilize external contextual knowledge, particularly when such external information conflicts with the model's inherent parametric

---

*Corresponding authors.

39th Conference on Neural Information Processing Systems (NeurIPS 2025).

knowledge (which may be outdated or inaccurate)? Researchers have investigated this question through various approaches, including negation injection [11, 20] and entity substitution [23, 47], revealing that LLMs often stubbornly adhere to their original beliefs. Other studies [39, 16] have demonstrated that the fluency of external context and the strength of the model's prior beliefs significantly influence LLMs' receptiveness to external knowledge. While these studies have revealed behavioral characteristics of LLMs in knowledge conflict scenarios, our understanding of the underlying mechanisms remains limited. Research into this question is not only crucial for designing superior context injection strategies and enhancing models' ability to integrate external knowledge but also provides theoretical guidance for building more reliable language models, advancing the widespread application of retrieval-augmented LLMs.

To unravel these mechanisms, we propose entity-aware probing – a framework that traces knowledge propagation as entity flows through transformer layers. Traditional probes fail in dynamic settings due to fixed label constraints. We overcome this by introducing special token markers combined with low-rank adaptation (LoRA), enabling ad-hoc entity tracing without altering base model parameters. This allows us to quantify how contextual and parametric knowledge evolve across layers during conflict scenarios. Our analysis reveals three key insights: (1) *Specialized routing*: Contextual and parametric knowledge are processed by distinct attention heads, with minimal cross-type competition. (2) *Asymmetric dynamics*: Contextual knowledge emerges abruptly in early layers, while parametric knowledge accumulates gradually via MLPs. (3) *Superpositional reconciliation*: Conflicting knowledge persists in residual streams, but aligned signals from multiple sources reinforce each other's magnitude, directly influencing output selection.

**Our contributions**. Our main contributions are twofold: (1) An entity-aware probing framework for dynamic knowledge tracing — enables ad-hoc entity relevance detection without modifying base model parameters. (2) Empirical insights into RAG dynamics - offers a foundation for future studies on knowledge interaction mechanisms.

## 2 Preliminaries

To understand how *parametric knowledge* and *contextual knowledge* reconcile within the model, we first reformulate the standard transformer architecture [35] from a residual stream perspective [7, 26] (Section 2.1). This reformulation allows us to isolate and analyze the information written into the residual stream by each component (MLP, attention, and individual attention heads), providing a foundation for studying their functional roles and actual knowledge propagation dynamics. We then introduce probing techniques (Section 2.2) as diagnostic tools to empirically study how models dynamically reconcile conflicting knowledge sources during this accumulation process.

### 2.1 Residual stream analysis of transformer computation

Standard transformer models process tokens through a stack of identical layers, each containing self-attention and feed-forward sublayers. We begin by equivalently rewriting this computation as cumulative updates to a residual stream. Specifically, in an $L$-layer decoder-only transformer, the hidden state of $t$-th token evolves as follows:

$$h_l^{(t)} = h_{l-1}^{(t)} + \Delta h_l^{\text{attn}}(t) + \Delta h_l^{\text{mlp}}(t), \tag{1}$$

where $\Delta h_l^{\text{attn}}(t)$ and $\Delta h_l^{\text{mlp}}(t)$ denote the contributions written to the residual stream by the self-attention and MLP sublayers, respectively. In retrieval-augmented generation, externally retrieved knowledge is introduced into the model's internal processing stream, where these sublayers form separate pathways for knowledge propagation and serve complementary roles.

**Parametric knowledge access**. The emergence of *parametric knowledge* has been observed to strongly correlate with MLP sublayer outputs [10, 25]. Specifically, denoting the MLP layer input as $\hat{h}_l^{(t)}$, the parametric knowledge access can be viewed as querying non-linear key-value memories:

$$\Delta h_l^{\text{mlp}}(t) = W_{\text{down}} \cdot \sigma(W_{\text{up}} \cdot \hat{h}_l^{(t)}), \tag{2}$$

where the input $\hat{h}_l^{(t)}$ is first mapped to key space through matrix $W_{\text{up}}$ and non-linear activation function $\sigma$. Subsequently, the corresponding token-specific knowledge fragments $\Delta h_l^{\text{mlp}}(t)$ are

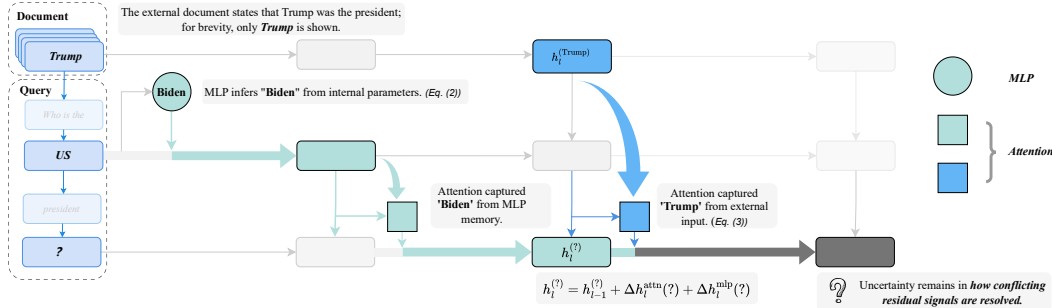

Figure 1: Schematic illustration of a knowledge conflict scenario: The external document describes 'Trump' as the U.S. President, whereas the model parameters encode 'Biden' as the U.S. President.

queried through $W_{\text{down}}$ and injected into the residual stream. This is a simplified description of parametric knowledge access; for more detailed modeling of memories querying, we refer readers to Geva et al. [10], Meng et al. [25]. We provide an illustrative example in Figure 1, where for a query "Who is the US president?", the parametric knowledge 'Biden' gradually emerges in the residual stream of relevant tokens (e.g., 'US') through the injection of multiple MLP layers.

**Knowledge routing by attention**. The attention mechanism serves as a crucial pathway for transferring knowledge between tokens. The content written into the residual stream by a single attention head is given by:

$$\Delta h_l^{\text{attn},(k)}(t) = W_O^{(k)} \left[ \text{Softmax} \left( \frac{W_Q^{(k)}(h_{l-1}^{(t)}) W_K^{(k)}(h_{l-1}^{1:t})^\top}{\sqrt{d_k}} \right) W_V^{(k)}(h_{l-1}^{1:t}) \right], \qquad (3)$$

where the four parameter matrices $W_{O,Q,K,V}^{(k)}$ of the $k$-th attention head determine its attention preferences. Specialized heads (e.g., retrieval heads [27, 38] that attend to external contexts and memory heads [17] accessing parametric knowledge) are found to exhibit differentiated sensitivity to these dual knowledge sources. The outputs of N attention heads are summed to form the attention sublayer output $\Delta h_l^{\text{attn}}(t) = \sum_{k=0}^{N} \Delta h_l^{\text{attn},(k)}(t)$. Taking Figure 1 as an example, both 'Trump' mentioned in the context and 'Biden' encoded in the model parameters influence the model's response to the query "Who is the US president?" through the attention sublayer output $\Delta h_l^{\text{attn}}(t)$.

Based on this formulation, we analyze the model's behavioral mechanisms under the RAG setting by probing the accumulated information in each layer's residual stream $h_l^{(t)}$, MLP output $\Delta h_l^{\text{mlp}}(t)$, attention layer output $\Delta h_l^{\text{attn}}(t)$, and individual attention head outputs $\Delta h_l^{\text{attn},(k)}(t)$.

## 2.2 Probing as mechanistic diagnostics

Probing techniques [1, 2, 12, 26] serve as diagnostic tools to analyze how linguistic properties and knowledge components are dynamically encoded in neural representations. A probe $f_\theta : \mathbb{R}^{d_{\text{model}}} \to \mathcal{Y}$ typically implemented as a lightweight classifier, is trained to decode target properties $y \in \mathcal{Y}$ (e.g., POS tags [34], named entities [15], or knowledge veracity labels [28]) directly from the residual stream, without modifying the base model's parameters.

**Limitations of traditional probing**. The classification head $f_\theta$ of traditional probes is static—it is trained to recognize only a fixed set of labels $\mathcal{Y}$ (e.g., predefined POS tags or entity types) during training. This rigidity makes it brittle when handling *ad-hoc* probing targets at test time. For example, to probe whether a residual stream contains information about a specific entity, you cannot feasibly train an entity-specific probe for every possible entity.

# 3 Probing framework for knowledge conflict internal mechanisms

## 3.1 Competing hypotheses on knowledge conflict

Given a query $q$ (e.g., "Who is the current U.S. President?") and an external document $\mathcal{D}$ returned by retrievers, we define *parametric knowledge* as all entities encoded within the model parameters that are objectively relevant to answering $q$, while *contextual knowledge* comprises answer-relevant entities derived from $\mathcal{D}$. These knowledge sources may conflict—for instance, parametric knowledge might encode "Biden" whereas retrieved documents $\mathcal{D}$ might state "After January 2025, Trump becomes the new U.S. President." We investigate how language models process and reconcile competing knowledge sources through their residual streams. Specifically, we propose two competing hypotheses about the model's internal mechanisms for handling knowledge conflicts:

- *Attention competition hypothesis*: Contextual and parametric knowledge compete for dominance through attention patterns, with the prevailing signal propagating through successive layers to determine the final output.
- *MLP arbitration hypothesis*: Both knowledge types are initially aggregated in the residual stream, with MLP layers subsequently injecting modulation signals that suppress conflicting information before final prediction.

## 3.2 Entity-aware probing architecture

To investigate these two hypotheses, we design an entity relevance probing task. Given an external context $\mathcal{D}$, a query $q$, and a specific entity $e$, the objective is to probe the base LLM's belief about "whether entity $e$ is relevant for answering query $q$." We follow the approach of Ye et al. [42] to resolve the limitations of traditional probing outlined in Section 2.2, adapting the probe $f_\theta$ for entity relevance probing.

Specifically, as illustrated in Figure 2, we append a triple "`[START] {target_entity} [END]`" after the external context $\mathcal{D}$ and query $q$. Here, `[START]` and `[END]` are reserved special tokens that explicitly indicate the target entity to the probe. To enable the model to process this structured input while preserving inherent processing mechanisms unchanged, we introduce a *small* trainable rank-8 LoRA update [13] exclusively to the input embedding layer. The base LLM parameters remain **frozen** during fine-tuning, with only the linear classification head and LoRA components updated to predict entity relevance at the `[END]` token position. This triple and LoRA update can be viewed as a form of "soft prompt" designed to elicit the model's conflict reconciliation behavior. For a discussion of the intuition behind this approach and its limitations, please refer to Appendix B.

To validate that the entity relevance predictions primarily emerge from the model's inherent knowledge processing mechanisms rather than the introduced lightweight adaptation, we conduct an ablation study comparing LoRA-enhanced probes on both instruct and randomly initialized LLM variants. As shown in Table 1, the sharp performance gap between *Probing* and *Probing(Random)* demonstrates that the predictive patterns predominantly originate from knowledge pathways encoded in their parameters, with the LoRA update serving merely as an interface for observation rather than introducing substantive functionality. These findings align with the results of Ye et al. [42].

## 3.3 Probe training data construction

We construct probe training data by adapting the HotpotQA-distractor dataset [41], leveraging its sentence-level human-annotated evidence to precisely identify entities required for answering queries. Each sample contains a multi-hop query $q$ and an external document $\mathcal{D}$ composed of 10 independent passages, where exactly two passages contain answer evidence while the remaining eight serve as distractors. We provide the two evidence passages and query $q$ as input to `Deepseek-v3-0324`, prompting it to generate a step-by-step answer to $q$. We then prompt the model to explicitly list entities involved in its generated answer (prompt template in Table 3). Entities appearing in both the evidence sentences and the model's answer are labeled as necessary (Yes). We then extract all entities mentioned in $\mathcal{D}$ using the prompt in Table 4, designating those not identified as necessary as irrelevant (No). From 6,200 processed HotpotQA samples, each generates training instances $s_i = <\mathcal{D}_i, q_i, \{e_i^{<j>}, y_i^{<j>}\}_{j=1}^N>$, where $y_i^{<j>} \in \{\text{Yes}, \text{No}\}$ indicates whether entity $e_i^{<j>}$ is relevant to answering $q_i$.

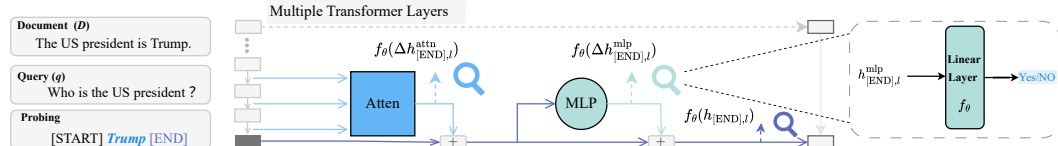

Figure 2: Schematic illustration of the knowledge dynamics tracing formulation: We probe the information of a specified entity (in this case, Trump) at three critical computation points within each Transformer layer: Attention Update, MLP Update, and Layer Output.

### 3.4 Knowledge dynamics tracing formulation

Let $f_\theta(h_{[END]}) : \mathbb{R}^{d_{model}} \to \mathcal{Y}$ denote the probe constructed in entity relevance probing task, where $\mathcal{Y} \in \{\text{Yes}, \text{No}\}$ represents the model's belief about whether entity $e$ is relevant for answering query $q$. To test the competing hypotheses outlined in Section 3.1, we systematically analyze the evolution of entity relevance beliefs by applying our probes at three critical points within each transformer layer's computation:

- *Attention-update entity probing*: $f_\theta(\Delta h_{[END],l}^{attn})$ reveals the direct impact of attention mechanisms on model beliefs, where the attention module routes both knowledge types through the preceding tokens' KV cache.

- *MLP-update entity probing*: $f_\theta(\Delta h_{[END],l}^{mlp})$ identifies the MLP module's direct influence on belief formation, corresponding to the model's integration of conflicting knowledge sources.

- *Layer output entity probing*: $f_\theta(h_{[END],l})$ tracks how entity $e$'s information aggregates across layers through successive attention and MLP updates up to layer $l$.

## 4 Probe validation via attribution task

To validate whether our entity-aware probe accurately reflects the model's internal beliefs about entity relevance, we conduct a controlled attribution task that evaluates two critical properties: *(i)* whether the probe identifies entities truly relevant to query $q$ within retrieved documents $\mathcal{D}$, and *(ii)* whether predictions derive from the base model's knowledge processing rather than the probe's parametric capacity.

### 4.1 Experimental setup

**Task definition**: Given a query $q$ and an external document $\mathcal{D}$ containing $N$ independent passages $\{p_1, ..., p_N\}$, the *passage-level attribution* task requires identifying passages containing answer evidence. For *sentence-level attribution*, we further pinpoint specific evidence-bearing sentences within each passage. We evaluate attribution performance on three established QA benchmarks—HotpotQA [41], TriviaQA [18], and SQuAD [30]—adapted for both passage-level and sentence-level attribution tasks. We measure attribution accuracy using standard information retrieval metrics: `Precision`, `Recall` and `F1`. Implementation details for constructing the attribution benchmarks and formal metric definitions appear in Appendix C. Please refer to Appendix G for examples of the dataset.

**Probing-based attribution**: We address the attribution task through entity relevance probing. For each passage $p_i$ (or sentence $s_j$) in document $\mathcal{D}$, we probe all mentioned entities $\mathcal{E}(p_i)$. Specifically, we apply the probe $f_\theta$ to evaluate relevance for each entity $e \in \mathcal{E}(p_i)$, where relevance is defined by:

$$\text{Relevant}(e) = \mathbb{I}\big[P_\theta(\text{Yes}|e, q, \mathcal{D}) > P_\theta(\text{No}|e, q, \mathcal{D})\big]. \tag{4}$$

We mark the passage/sentence as an attribution target if it contains *any* entity with $\text{Relevant}(e) = 1$.

**Compared baselines**: Three critical comparisons are established as follows: *(1) Prompting*: Instructs the base LLM to directly output attribution results through one-shot manual demonstrations, representing the simplest way to test its internal beliefs. The prompts we used are shown in Table 5. *(2) Prompting(CoT)*: Builds upon (1) by incorporating entity information extracted by the base LLM into the prompt and adding a CoT-style instruction that guides the model to reason through

Table 1: Citation recall (R), precision (P) and F1 (F1) are standard metrics for evaluating attribution. Avg. denotes the mean F1 score across all datasets, reflecting overall model or method performance.

| Model | | Attribution Method | TriviaQA | | | SQuAD | | | HotpotQA | | | Avg. |
|---|---|---|---|---|---|---|---|---|---|---|---|---|
| | | | R | P | F1 | R | P | F1 | R | P | F1 | |
| Proprietary Models | GPT-4o | *Prompting* | 93.0 | 92.5 | 92.7 | 99.5 | 99.5 | 99.5 | 76.3 | 82.6 | 76.9 | 89.7 |
| | Deepseek-v3-0324 | *Prompting* | 92.0 | 90.8 | 91.2 | 100.0 | 100.0 | 100.0 | 77.1 | 81.4 | 76.1 | 89.1 |
| | GLM-4-plus | *Prompting* | 79.0 | 62.3 | 66.0 | 100.0 | 99.4 | 99.6 | 76.6 | 74.6 | 72.2 | 79.3 |
| Open-source Models | Qwen2.5-7B | *Prompting* | 36.5 | 33.5 | 34.4 | 92.5 | 92.3 | 92.3 | 45.9 | 61.1 | 49.7 | 58.8 |
| | | *Prompting(CoT)* | 72.0 | 65.0 | 67.1 | 96.0 | 94.9 | 95.3 | 62.8 | 73.2 | 64.7 | 75.7 |
| | | *Probing* | 83.6 | 74.0 | 78.5 | 87.4 | 93.8 | 90.5 | 48.9 | 51.3 | 50.0 | 73.0 |
| | | *Probing(Random)* | 64.2 | 20.2 | 30.8 | 88.8 | 10.2 | 18.0 | 38.2 | 7.7 | 12.8 | 20.3 |
| | Qwen3-8B | *Prompting* | 51.0 | 50.2 | 50.4 | 57.5 | 57.5 | 57.5 | 50.6 | 66.3 | 55.0 | 54.3 |
| | | *Prompting(CoT)* | 77.5 | 76.9 | 77.1 | 98.5 | 98.5 | 98.5 | 63.5 | 81.7 | 69.2 | 81.6 |
| | | *Probing* | 48.5 | 97.2 | 64.7 | 94.8 | 95.0 | 94.9 | 60.9 | 46.8 | 52.9 | 70.8 |
| | | *Probing(Random)* | 68.7 | 19.9 | 30.8 | 62.4 | 9.8 | 17.0 | 58.3 | 7.0 | 12.5 | 20.1 |
| | LLaMA3.1-8B | *Prompting* | 42.5 | 35.9 | 37.8 | 48.0 | 45.4 | 46.3 | 31.2 | 40.5 | 33.0 | 39.0 |
| | | *Prompting(CoT)* | 36.5 | 35.4 | 35.8 | 91.5 | 91.3 | 91.3 | 58.7 | 74.0 | 62.3 | 63.1 |
| | | *Probing* | 46.4 | 97.5 | 62.8 | 88.4 | 94.2 | 91.2 | 47.0 | 56.2 | 51.2 | 68.4 |
| | | *Probing(Random)* | 61.7 | 19.9 | 30.1 | 83.4 | 10.0 | 17.9 | 69.5 | 7.0 | 12.7 | 20.2 |

entity–citation relations before producing the final attribution. The prompts are shown in Table 6. *(3) Probing(Random)*: Reinitializes the base LLM's parameters randomly (with parameters frozen during probe training) while maintaining identical probing architecture, isolating the probe's capacity to learn relevance patterns independently of LLM's knowledge processing.

## 4.2 Results analysis

As shown in Table 1, we evaluate mainstream open-source models (Qwen2.5, Qwen3, LLaMA3.1) and state-of-the-art proprietary models on the attribution task. Our analysis reveals two key findings:

**LLMs possess intrinsic beliefs about entity relevance**: For strong proprietary models, the simple *Prompting* method alone suffice to elicit accurate entity relevance judgments. Although smaller open-source models initially perform poorly, their entity relevance judgment significantly improves after more refined prompt design (*Prompting → Prompting(CoT)*), approaching or even surpassing some proprietary models. This suggests that these models internally possess knowledge of entity relevance, but this belief may require appropriate elicitation methods. The *Probing* approach achieves performance comparable to Prompting(CoT), indicating that probing serves as an alternative elicitation method that can effectively extract entity relevance information directly from hidden states.

**Probing results stem from the LLM's own beliefs rather than probe parameters**: Our critical baseline – probing a randomly reinitialized LLM – demonstrates that attribution performance collapses to near-random levels (average F1 of 20.3 vs. 73.0 for intact Qwen2.5). Precision metrics plummet to 10.2 on SQuAD and 7.7 on HotpotQA, comparable to random selection from candidate passages/sentences. This confirms that successful attribution depends on the base LLM's knowledge processing mechanisms rather than the probe's parametric capacity.

## 5 Understanding knowledge reconciliation within LLMs

### 5.1 Experimental setup

**Evaluation Data**: To investigate LLMs' knowledge reconciliation mechanisms, we construct knowledge conflict scenarios based on the HotpotQA dataset. Since HotpotQA is derived from Wikipedia—the primary pretraining corpus for LLMs—we assume its factual descriptions align with LLMs' *parametric knowledge*. For each query, we leverage sentence-level evidence from HotpotQA and instruct `Deepseek-v3-0324` to replace original factual entities with fictional counterfactual entities, generating rewritten evidence and forming conflict-inducing external documents. Using prompt templates (Table 3), we create 786 samples, each formatted as $s_i = <\mathcal{D}_f, \mathcal{D}_{cf}, q, \mathcal{E}>$, where $\mathcal{D}_f$ denotes the original document from HotpotQA, $\mathcal{D}_{cf}$ denotes the counterfactual external document, $q$ denotes query, $\mathcal{E}$ denotes all mentioned entities, categorized into:



Figure 3: Information intensity (measured by PRAUC scores) across all attention heads in LLaMA 3.1. In the titles of subfigures (a-d), the left side of the '$\rightarrow$' indicates the type of external documents, where $\mathcal{D}_{cf}$ represents counterfactual documents and $\mathcal{D}_{cf} \cup \mathcal{D}_f$ represents the union of original and counterfactual documents. The right side of the arrow indicates the probing target, where $\mathcal{E}_{cf}$ represents counterfactual entities (aligned with contextual knowledge in $\mathcal{D}_{cf}$) and $\mathcal{E}_f$ represents factual entities (consistent with both parametric knowledge and contextual knowledge in $\mathcal{D}_f$).

- *parametric knowledge* ($\mathcal{E}_f$): Factual entities from $\mathcal{D}_f$ that align with the model's internal knowledge for answering $q$.

- *contextual knowledge* ($\mathcal{E}_{cf}$): Counterfactual entities from $\mathcal{D}_{cf}$ that conflict with parametric knowledge.

- *irrelevant knowledge* ($\mathcal{E}_{irr}$): Remaining entities in $\mathcal{E} - (\mathcal{E}_f \cup \mathcal{E}_{cf})$, unrelated to resolving $q$.

**Metrics**: We employ the PRAUC (Precision-Recall Area Under the Curve) [6] score to quantify the information intensity of parametric knowledge and contextual knowledge in residual stream. The metric robustly addresses class imbalance (relevant entities are rare compared to irrelevant ones). Specifically, query-relevant entities ($\mathcal{E}_f$ or $\mathcal{E}_{cf}$) are treated as the positive class, while unrelated entities ($\mathcal{E}_{irr}$) form the negative class. PRAUC is computed using the probe $f_\theta$'s entity relevance scores $P_\theta(\text{Yes}|e, q, \mathcal{D})$. A higher PRAUC score indicates that the probe can more significantly detect the corresponding knowledge at the specified probing position.

## 5.2 Attention head analysis

We analyze the information intensity of all attention heads in LLaMA3.1, Qwen2.5, and Qwen3.0 models using probes. This section corresponds to the 'Attention-update entity probing' discussed in Section 3.4 formulation. Figure 3 presents the experimental results for LLaMA3.1, while results for Qwen2.5 and Qwen3.0 are shown in Appendix Figures 7 and 8 due to space constraints. All three models exhibit the following consistent patterns:

**(i) Contextual and parametric knowledge are transferred by distinct attention heads**: When feeding counterfactual context $\mathcal{D}_{cf}$ (where contextual knowledge conflicts with the model's parametric knowledge) as external documents, subfigures (a) and (b) of Figure 3 measure the information intensity of contextual knowledge and parametric knowledge in individual attention heads, respectively. The results reveal that these two types of knowledge are routed through separate sets of attention heads, aligning with findings from existing mechanistic interpretability studies on attention heads [17, 27, 38]. Among the top-10 heads with the highest information intensity for each knowledge type, only 2 heads overlap in Qwen3.0, and 1 head overlaps in LLaMA3.1 and Qwen2.5. For clarity, we refer to these as contextual heads (routing contextual knowledge) and memory heads (routing parametric knowledge).

**(ii) Fundamental differences in transfer mechanisms between knowledge types**: Subfigures (a) and (b) of Figure 3 highlight a key distinction: the information intensity of contextual heads significantly exceeds that of memory heads. This suggests that contextual knowledge transfer resembles explicit retrieval, where *complete knowledge units* are captured and propagated by a small subset of attention heads. In contrast, parametric knowledge likely involves more intricate processing—each memory head may transfer *fragmented knowledge pieces*, which require multi-layer aggregation to form coherent knowledge units. This might explain why the information intensity of any single attention head remains relatively low. We provide further evidence supporting this hypothesis in Section 5.3.

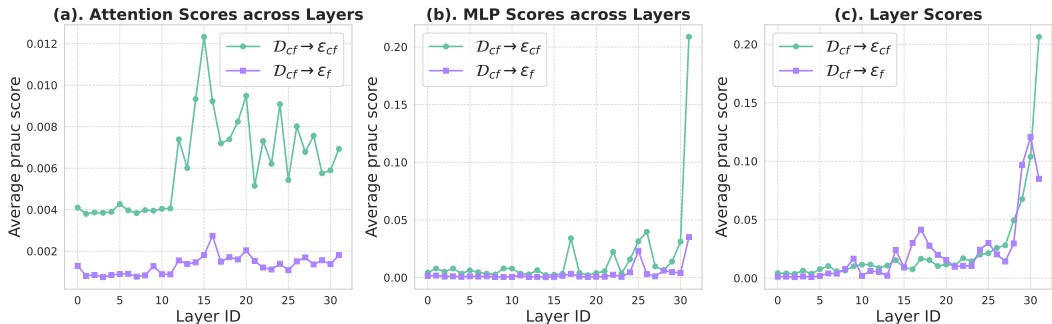

Figure 4: Information intensity probing results (measured by PRAUC scores) for LLaMA 3.1 with counterfactual document $\mathcal{D}_{cf}$ as context. $\mathcal{D}_{cf} \rightarrow \mathcal{E}_{cf}$ and $\mathcal{D}_{cf} \rightarrow \mathcal{E}_f$ represent the average information intensity of counterfactual entities and factual entities respectively.

**(iii) Competition between same-type knowledge, but not across types**: Subfigures (c) and (d) of Figure 3 present probing results under mixed contexts ($\mathcal{D}_f \cup \mathcal{D}_{cf}$), where factual (parametric-aligned) and counterfactual (conflicting) knowledge coexist. Here, contextual knowledge types (factual vs. counterfactual) compete for attention within contextual heads: the information intensity for both in subfigures (c) and (d) is substantially lower than in subfigure (a). However, the presence of factual knowledge in the context has minimal impact on memory heads (parametric knowledge routing). For LLaMA3.1 and Qwen3.0, the average information intensity of top-10 memory heads slightly increases ($\leq 5\%$), while Qwen2.5 shows a minor decrease ($\leq 5\%$). This indicates competition occurs only within the same knowledge type (contextual vs. contextual), not across distinct types (contextual vs. parametric).

## 5.3 Residual stream analysis

Building on the probing framework (Section 3.4), we analyze the information intensity of parametric knowledge ($\mathcal{E}_f$) and contextual knowledge ($\mathcal{E}_{cf}$) across transformer layers by probing Attention updates, MLP updates, and layer outputs. Key findings from experiments on LLaMA 3.1 under counterfactual documents $\mathcal{D}_{cf}$ and mixed documents $\mathcal{D}_{cf} \cup \mathcal{D}_f$ (Figures 4-5, with Qwen2.5/3.0 results in Appendix Figures 9-12) reveal:

**(i) Divergent propagation mechanisms for parametric vs. contextual knowledge**:

- *Parametric knowledge* ($\mathcal{E}_f$) exhibits low information intensity in early-layer attention and MLP updates but shows rapid amplification in later layers (Figure 4). This suggests fragmented parametric knowledge is gradually consolidated through multi-layer interactions, forming coherent units only near the output layer.
- *Contextual knowledge* ($\mathcal{E}_{cf}$), in contrast, is immediately detectable in early attention updates, persists latently in residual streams, and is reactivated in later stages for answer generation. This reflects its direct encoding from external contexts.

**(ii) Superposition as a Reconciliation Pathway**: When replacing $\mathcal{D}_{cf}$ with a mixed document $\mathcal{D}_{cf} \cup \mathcal{D}_f$ (Figure 5), parametric knowledge ($\mathcal{E}_f$) gains competitive advantage in output layers compared to scenarios without contextual support (Figure 4 (c)). This indicates that:

(1) Factual knowledge from multiple sources (parametric + contextual) can superpositionally reinforce their information intensity in the residual stream, enhancing their influence on final outputs.

(2) Conflicting knowledge (e.g., $\mathcal{E}_{cf}$) coexists in the residual stream without erasure, suggesting LLMs preserve opposing evidence rather than discarding it.

## 5.4 Evidence from Native Generation Dynamics

Although we aim to minimize interference, probing may inadvertently influence the model's generation behavior. To further validate the knowledge reconciliation mechanisms identified through probing, we conduct two intervention experiments using LLaMA3.1 in its native generation setting.

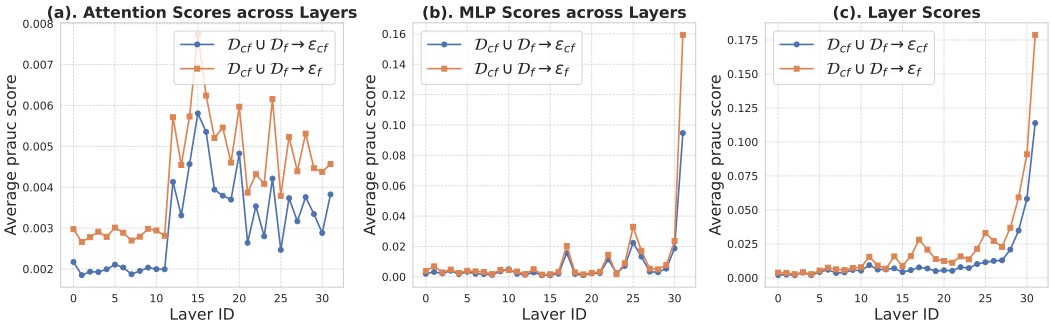

Figure 5: Information intensity probing results (measured by PRAUC scores) for LLaMA 3.1 using the union of original and counterfactual documents $\mathcal{D}_{cf} \cup \mathcal{D}_f$ as context. $\mathcal{D}_{cf} \cup \mathcal{D}_f \rightarrow \mathcal{E}_{cf}$ and $\mathcal{D}_{cf} \cup \mathcal{D}_f \rightarrow \mathcal{E}_f$ represent the average information intensity of counterfactual entities and factual entities respectively.

Table 2: Causal impact of zeroing top-10 context heads and memory heads on knowledge reconciliation behavior under native generation mode. The proportions of the `don't know` and `other` options are combined under **Other** column in the table.

| Experimental Setting | Factual Answer | Counterfactual Answer | Other |
|---|---|---|---|
| *Original Model* | 22.70% | 54.60% | 22.70% |
| w/o Context Heads | **34.90%** | 43.60% | 21.50% |
| w/o Memory Heads | 9.20% | **65.50%** | 25.30% |

For ease of evaluation, we formulate the test in a multiple-choice format, where `factual_answer`, `counterfactual_answer`, `don't know`, and `other` are randomly assigned to options A, B, C, D.

**(1) Causal Intervention via Zeroing Attention Heads.** Given a counterfactual document as context, we zero out the identified top-10 *context heads* and *memory heads* to observe the impact of this intervention on LLM response generation. The prompt used is shown in Table 10. As shown in Table 2, removing context heads reduces contextual influence (54.6% → 43.6%) and increases reliance on parametric knowledge (22.7% → 34.9%). Conversely, removing memory heads weakens parametric influence (22.7% → 9.2%) and strengthens contextual responses (54.6% → 65.5%). These results provide causal evidence that the identified heads perform the reconciliation function, as described in Section 5.2.

**(2) Attention Redistribution via Prompt Manipulation.** We design three different prompts to enhance the model's attention to counterfactual documents in the context: *Duplicate and append*, *Insert prompt randomly*, and *Tag and append*. Detailed descriptions of these strategies are provided in Appendix H. In the left panel of Figure 6, we provide both factual and counterfactual documents as context to the model. After applying the three strategies above, the proportion of `counterfactual_answers` increase significantly, from 27.7% to 45.3%, 40.9%, and 41.2%, respectively. This indicates that there exists competition between `factual_answer` and `counterfactual_answer` from the context, where strengthening one weakens the other. In the right panel of Figure 6, the context contains only counterfactual documents. After applying the three strategies above, the proportion of `counterfactual_answer` show no significant change. This suggests that there is no direct competition between `counterfactual_answer` from the context and `factual_answer` from the model's parameters; strengthening one does not notably affect the other.

## 5.5 Discussions

Our experimental findings reveal nuanced interactions between the two hypothesized mechanisms in Section 3.1, suggesting a hybrid processing strategy rather than exclusive adherence to either hypothesis.

**Attention Competition Hypothesis Revisited**. The hypothesis posits that contextual and parametric knowledge compete through attention patterns. Our results partially support this view but with critical

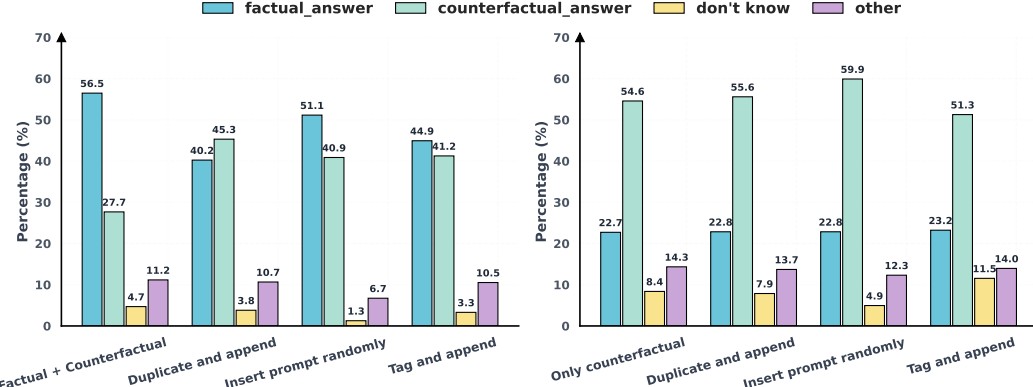

Figure 6: Knowledge competition between same-type knowledge (left panel), but not across types (right panel). "Factual+Counterfactual" indicates using both factual and counterfactual documents as context, while "Only counterfactual" refers to using only counterfactual documents as context. Three strategies are (see Appendix H for detailed descriptions) designed to direct the LLM's attention toward counterfactual documents.

qualifications: *(1) Distinct routing pathways*: The separation of contextual vs. parametric knowledge into dedicated attention heads (Section 5.2(i)) supports the notion of parallel processing rather than direct competition. This suggests attention mechanisms implement a "division of labor" strategy rather than winner-takes-all competition. *(2) Within-type competition*: The observed suppression between factual/counterfactual contextual knowledge in mixed contexts (Section 5.2(iii)) aligns with attention-level competition, but only within the same knowledge type. This implies attention heads implement *intra-type* competition rather than cross-type rivalry.

**MLP Arbitration Hypothesis Revisited**. The hypothesis suggests MLP layers resolve conflicts through suppression of contradictory signals. Our analysis reveals more complex dynamics: *(1) Persistence of Conflicting Signals*: Despite MLP operations on the residual stream, conflicting contextual and parametric knowledge maintain a residual presence across layers (Section 5.3(i)), contradicting the suppression hypothesis. *(2) Superpositional reconciliation*: The enhanced parametric knowledge intensity under mixed contexts (Section 5.3(ii)) indicates MLPs may implement additive superposition of aligned evidence from multiple sources, with the accumulation magnitude directly determining its influence on final outputs.

## 6 Conclusions

This study presents a mechanistic understanding of how LLMs reconcile conflicting parametric and contextual knowledge in RAG frameworks. By introducing an entity-aware probing methodology, we uncover three fundamental principles governing knowledge dynamics: (1) Attention head specialization enables parallel processing of distinct knowledge types through dedicated routing pathways; (2) Asymmetric propagation mechanisms reveal contextual knowledge emerges abruptly via attention while parametric knowledge accumulates gradually through MLP layers; and (3) Superpositional reconciliation demonstrates that conflicting knowledge persists in residual streams, with aligned multi-source signals reinforcing each other to determine output selection. This work provides both theoretical foundations and practical guidance for building more reliable and interpretable retrieval-augmented language models.

## Acknowledgements

The authors wish to thank the anonymous reviewers for their helpful comments. This work was partially funded by National Natural Science Foundation of China (No.62476061, 62441602, 62206057), Shanghai Rising-Star Program (23QA1400200), and Natural Science Foundation of Shanghai (23ZR1403500).

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

## Technical Appendices and Supplementary Material

## Limitations

Our study has several limitations that warrant careful consideration. First, while our entity-aware probing framework demonstrates effectiveness in tracing knowledge flows within decoder-only transformers like LLaMA and Qwen series, its generalizability to other architectures (e.g., encoder-decoder models) or significantly larger models remains untested. Second, our experiments use synthetic counterfactual documents constructed via LLM-based entity substitution. While this ensures controlled evaluation, real-world retrieval contexts may contain more subtle conflicts involving temporal reasoning, numerical discrepancies, or conflicting logical implications—not merely entity-level replacements. Future work should validate our findings on naturally occurring knowledge conflicts from dynamic knowledge sources.

## A  Related work

**Retrieval-augmented generation and knowledge conflicts** Large language models often face challenges such as outdated knowledge and hallucinations [33]. Retrieval-Augmented Generation (RAG) offers a solution to these issues by retrieving and integrating external data as a supplement [46, 19, 14]. External knowledge introduces challenges, especially conflicts with model-parametric knowledge [40, 31]. Such conflicts introduce source-level uncertainty and may significantly degrade model performance [40, 29]. While some studies suggest a model preference for parametric memory [24], others highlight a greater reliance on external knowledge [4]. Naturally, various methods have been proposed to mitigate these conflicts [36, 3, 22, 9]. However, how models handle such conflicts remains an area for further research.

**Internal Probing for Model Interpretability** A common approach to interpretability is probing model internals [44, 37, 5, 43]. Yuksekgonul et al. [44], Wu et al. [37] investigated the behavior of attention mechanisms when the model is generating factual answers. There are also approaches known as lens methods. These include LL-based methods [26] that analyze information flow within the attention mechanism [32], and approaches that incorporate Logit Lens during backpropagation [21]. We effectively improve upon these probing methods and adapt them to QA tasks, showing that they yield both promising performance and effective interpretability.

## B  Intuition and Limitations of Entity-aware Probing

First, we consider a simple case: **Document**: *The capital of France is London.* **Q**: *Where is the capital of France?* **A**: *The capital of France is ___?* In this case, the candidate entity tokens include "London" (contextual knowledge) and "Paris" (parametric knowledge). Here, the answer prefix "The capital of France is" essentially serves to elicit the model's conflict reconciliation behavior. In standard generation, not every token prediction involves knowledge-intensive reasoning or conflict resolution. Many tokens (e.g., articles, conjunctions, formatting) are generated through relatively shallow pattern matching. The LoRA updates and the entity triples appended after the document and question in our work can essentially be viewed as a trainable soft prompt. It serves a similar function to "The capital of France is" in eliciting the model's conflict reconciliation behavior. However, we must acknowledge that this approach increases the risk of altering the model's internal behavioral patterns. We believe that constructing more flexible probing methods with minimal intervention in model behavioral patterns represents an important open problem worthy of continued exploration.

## C  Attribution dataset adaptation and evaluation metrics

**Dataset adaptation**: We build our evaluation using three established QA benchmarks, adapting them to support passage- and sentence-level attribution tasks. HotpotQA serves as both the training set of the probe and a sentence-level attribution testing set (see Section 3.3 for training details), while SQuAD and TriviaQA are exclusively used for passage-level attribution evaluation (without dataset-specific probe fine-tuning).

- **Hotpotqa** [41]: Designed for multi-hop reasoning, HotpotQA provides sentence-level attribution target through human-annotated evidence sentences. Each query is paired with 10 passages, of which exactly 2 contain gold evidence sentences. The remaining 8 passages act as natural distractors, enabling fine-grained evaluation of sentence attribution without requiring synthetic negative sampling.
- **TriviaQA** [18]: For passage-level evaluation, we construct input documents by combining the gold passage from each test sample with 4 randomly selected passages from other test instances.
- **SQuAD** [30]: Following a similar passage-level evaluation protocol to TriviaQA but with shorter passages, we aggregate each gold passage with 9 randomly sampled irrelevant passages to form evaluation documents.

**Evaluation metrics**: We evaluate the results using three standard metrics: `Precision`, `Recall` and `F1`. Given that the ground truth passage/sentence indices are known, we compute the metrics as follows:

- `Precision` measures the proportion of correctly identified relevant passages among all predicted as relevant, computed as $|\text{Predicted} \cap \text{GroundTruth}|/|\text{Predicted}|$.
- `Recall` measures the proportion of correctly identified relevant passages among all truly relevant ones, computed as $|\text{Predicted} \cap \text{GroundTruth}|/|\text{GroundTruth}|$.
- `F1 score` is the harmonic mean of precision and recall, calculated as $(2 \cdot \text{Precision} \cdot \text{Recall})/(\text{Precision} + \text{Recall})$.

## D   Implementation Details

The experiments were conducted using the Openrlhf framework with DeepSpeed for distributed training. The model was trained with BF16 mixed precision and Flash Attention optimization to accelerate computation and reduce memory overhead. All model parameters frozen except for the LoRA adapter injected into the input embedding layer. Training utilized a global batch size of 256 with a micro-batch size of 4 for memory-efficient gradient accumulation. The AdamW optimizer was configured with a learning rate of 1e-5 and ZeRO Stage 2 optimization. The model was trained for 2 epochs. The input sequence length was capped at 3,000 tokens. Training the probe requires approximately 3 days on 8 A100 GPUs, and testing on a single dataset takes about 1 hour.

## E   Visualization of PRAUC Scores for Qwen2.5 and Qwen3



Figure 7: Information intensity (measured by `PRAUC` scores) across all attention heads in Qwen 2.5. In the titles of subfigures (a-d), the left side of the '→' indicates the type of external documents, where $\mathcal{D}_{cf}$ represents counterfactual documents and $\mathcal{D}_{cf} \cup \mathcal{D}_f$ represents the union of original and counterfactual documents. The right side of the arrow indicates the probing target, where $\mathcal{E}_{cf}$ represents counterfactual entities (aligned with contextual knowledge in $\mathcal{D}_{cf}$) and $\mathcal{E}_f$ represents factual entities (consistent with both parametric knowledge and contextual knowledge in $\mathcal{D}_f$).

## F   Full List of Prompts Used in the Experiments



Figure 8: Information intensity (measured by PRAUC scores) across all attention heads in Qwen 3.0. In the titles of subfigures (a-d), the left side of the '→' indicates the type of external documents, where $\mathcal{D}_{cf}$ represents counterfactual documents and $\mathcal{D}_{cf} \cup \mathcal{D}_f$ represents the union of original and counterfactual documents. The right side of the arrow indicates the probing target, where $\mathcal{E}_{cf}$ represents counterfactual entities (aligned with contextual knowledge in $\mathcal{D}_{cf}$) and $\mathcal{E}_f$ represents factual entities (consistent with both parametric knowledge and contextual knowledge in $\mathcal{D}_f$).

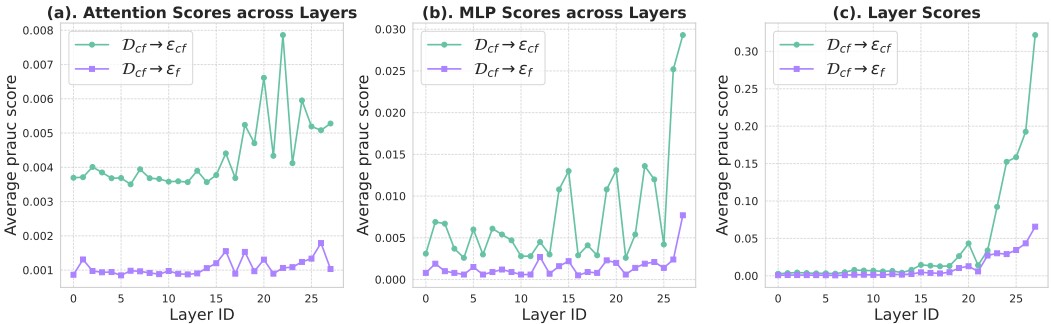

Figure 9: Information intensity probing results (measured by PRAUC scores) for Qwen 2.5 model with counterfactual document $\mathcal{D}_{cf}$ as context. $\mathcal{D}_{cf} \to \mathcal{E}_{cf}$ and $\mathcal{D}_{cf} \to \mathcal{E}_f$ represent the average information intensity of counterfactual entities and factual entities respectively.

Table 3: Prompt template for data generation.

---

**Example 1:**

Passage A: The 2015 Diamond Head Classic was a college basketball tournament ... Buddy Hield was named the tournament's MVP.
Passage B: Chavano Rainier "Buddy" Hield is a Bahamian professional basketball player for the Sacramento Kings of the NBA...

---

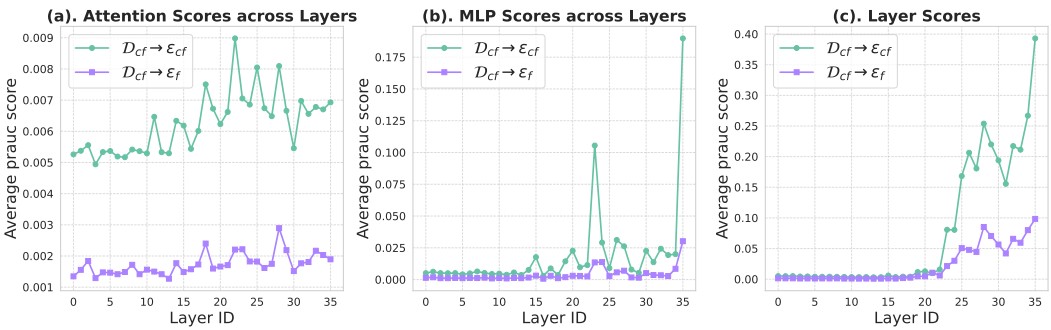

Figure 10: Information intensity probing results (measured by PRAUC scores) for Qwen 3.0 model with counterfactual document $\mathcal{D}_{cf}$ as context. $\mathcal{D}_{cf} \to \mathcal{E}_{cf}$ and $\mathcal{D}_{cf} \to \mathcal{E}_f$ represent the average information intensity of counterfactual entities and factual entities respectively.

Question: Which team does the player named 2015 Diamond Head Classic's MVP play for?
Answer: Sacramento Kings

Analysis:
   Step 1: Using the clue "2015 Diamond Head Classic," identify in Passage A that the MVP is "Buddy Hield."
   Step 2: Using the clue "Buddy Hield," find in Passage B that he currently plays for the "Sacramento Kings."
   Therefore, the answer is the Sacramento Kings.
   The entities involved in the aforementioned analysis, in logical order, are: ["2015 Diamond Head Classic", "Buddy Hield", "Sacramento Kings"]
   The entities that not reffered to in the questions are: ["Buddy Hield", "Sacramento Kings"].

Counterfactual Data Synthesis:
   Let's conduct a counterfactual analysis by replacing the entities that were mentioned in the analysis but did not appear in the Question.
   As we mentioned above, these entities are: ["Buddy Hield", "Sacramento Kings"]
   We will replace them with fictional entities: ["Alex Taylor", "Vancouver Bears"]
   Revised Passage A: The 2015 Diamond Head Classic was a college basketball tournament ... Alex Taylor was named the tournament's MVP.
   Revised Passage B: Alex Taylor is a Canadian professional basketball player for the Vancouver Bears of the NBA...
   Revised Question: Which team does the player named 2015 Diamond Head Classic's MVP play for?
   Revised Answer: Vancouver Bears
   Revised Analysis:
      Step 1: Using the clue "2015 Diamond Head Classic," identify in Passage A that the MVP is "Alex Taylor."
      Step 2: Using the clue "Alex Taylor," find in Passage B that he currently plays for the "Vancouver Bears."
      Therefore, the answer is the Vancouver Bears.
      The entities involved in the revised analysis, in logical order, are: ["2015 Diamond Head Classic", "Alex Taylor", "Vancouver Bears"]
      The entities that not reffered to in the questions are: ["Alex Taylor", "Vancouver Bears"]

**Example 2:**

Paragraph A: Several current and former members of the Pittsburgh Pirates – ... John Milner, Dave Parker, and Rod Scurry...
Paragraph B: David Gene Parker, nicknamed "The Cobra", is an American former player in Major League Baseball...
Question: Which former member of the Pittsburgh Pirates was nicknamed "The Cobra"?
Answer: David Gene Parker

Analysis
   Step 1: Using the clue "Pittsburgh Pirates" from the Question, identify in Paragraph A the list of current and former members that includes "John Milner," "Dave Parker," and "Rod Scurry."
   Step 2: Using the clue "nickname 'The Cobra'" from the Question, identify in Paragraph B that "David Gene Parker" is nicknamed "The Cobra."
   Step 3: Connect the information that "Dave Parker" from Paragraph A is the same individual as "David Gene Parker" from Paragraph B who is nicknamed "The Cobra."

Therefore, the answer is David Gene Parker.

Since 'David Gene Parker' and 'Dave Parker' refer to the same entity, we will use 'David Gene Parker' to refer to it in subsequent analysis.

The entities involved in the aforementioned analysis, in logical order, are: ["Pittsburgh Pirates", "David Gene Parker", "The Cobra"]

The entities that not reffered to in the questions are: ["David Gene Parker"]

```
Counterfactual Data Synthesis:
```

Let's conduct a counterfactual analysis by replacing the entities that were mentioned in the analysis but did not appear in the Question.

As we mentioned above, these entities are: ["David Gene Parker"]

We will replace them with fictional entities: ["Alexander T. Taylor"]

`Revised Paragraph A`: Several current and former members of the Pittsburgh Pirates – ... John Milner, Alex Taylor, and Rod Scurry...

`Revised Paragraph B`: Alexander T. Taylor, nicknamed "The Cobra," is an American former player in Major League Baseball...

`Revised Question`: Which former member of the Pittsburgh Pirates was nicknamed "The Cobra"?

`Revised Answer`: Alexander T. Taylor

`Revised Analysis`:

Step 1: Using the clue "Pittsburgh Pirates" from the Question, identify in Revised Paragraph A the list of current and former members that includes "John Milner," "Alex Taylor," and "Rod Scurry."

Step 2: Using the clue "nickname 'The Cobra'" from the Question, identify in Revised Paragraph B that "Alexander T. Taylor" is nicknamed "The Cobra."

Step 3: Connect the information that "Alex Taylor" from Paragraph A is the same individual as "Alexander T. Taylor" from Paragraph B who is nicknamed "The Cobra." Therefore, the answer is Alexander T. Taylor.

Entities involved in the revised analysis, in logical order, are: ["Pittsburgh Pirates", "Alexander T. Taylor", "The Cobra"]

Entities that are not referred to in the questions: ["Alexander T. Taylor"]

```
You are a logical analysis expert. Please provide your analysis and
counterfactual data synthesis of the following material in the manner
described above. In your analysis, the entities should be presented in
the form of a list.
Passage A: {Passage_Content}
Passage B: {Passage_Content}
Question: {Question_Content}
Answer: {Answer_Content}
Analysis:
```

Table 4: Prompt template for extracting entities from a sentence.

**Example 1:**
```
Sentence:  The thank you letter is an opportunity for you to remind the
interviewer why you are the best fit for the position and to address any
remaining issues.
Answer:  [thank you letter, interviewer, best fit, position, remaining
issues]
```

**Analysis:** The answer in every example is a list that contains the entities in the sentence. Entities refer to specific nouns or noun phrases with clear meanings that play a key role in the sentence. Therefore, these words are extracted as entities.

You are a language expert. Now, I will give you a sentence, and you need to output an entity list for the sentence in the format similar to the example. If there are no entities, return an empty list.

```
<sentence>
{Sentence}
</sentence>
```

Table 5: Prompt template for citation selection in reading comprehension tasks.

You are an expert in reading comprehension. You will be given a passage and a question, where the sentences or paragraphs in the passage are segmented by identifiers such as <c0>. Please read the document and question carefully, analyze which sentences or paragraphs need to be cited to answer this question, and then return a list of citations in the format: `<cite>[citation index only numbers]</cite>`.

**Example:**
```
[Document Start]
<C0>In the last two decades, several hundred planets have been detected
beyond our Solar-system.
<C1>Most of these extra-solar planets orbit sun-like stars.
<C2>A small number have been detected around stars that are in their
late evolutionary state.
<C3>The phase directly after the RGB stage, the Horizontal Branch (HB),
however, is still unexplored;
<C4>Besides its evolutionary stage, a star's chemical composition
appears to be a major indicator of its probability.
[Document End]

[Question]
How does the metallicity of PSR B1620 b, at around 1% of the Sun's,
compare to other known exoplanet host stars?

[Remind]
```
Please read the document and question carefully, analyze which sentences or paragraphs need to be cited to answer this question, and then return a list of citations in the format: `<cite>[citation index only numbers]</cite>`.

```
[Answer Citation List]
<cite>[0,3]</cite>
```

**Now get ready to handle the following test case.**

```
[Document Start]
<context>
[Document End]

[Question]
<question>

[Remind]
```
Please read the document and question carefully, analyze which sentences or paragraphs need to be cited to answer this question, and then return a list of citations in the format: `<cite>[citation index only numbers]</cite>`.

```
[Answer Citation List]
```

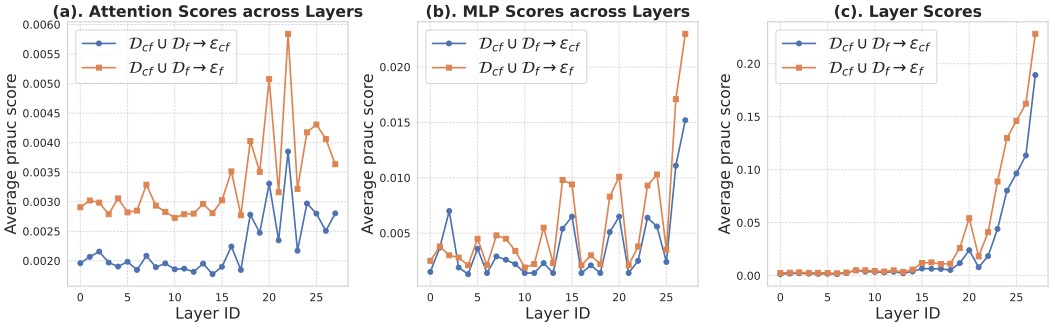

Figure 11: Information intensity probing results (measured by PRAUC scores) for Qwen 2.5 model using the union of original and counterfactual documents $\mathcal{D}_{cf} \cup \mathcal{D}_f$ as context. $\mathcal{D}_{cf} \cup \mathcal{D}_f \rightarrow \mathcal{E}_{cf}$ and $\mathcal{D}_{cf} \cup \mathcal{D}_f \rightarrow \mathcal{E}_f$ represent the average information intensity of counterfactual entities and factual entities respectively.

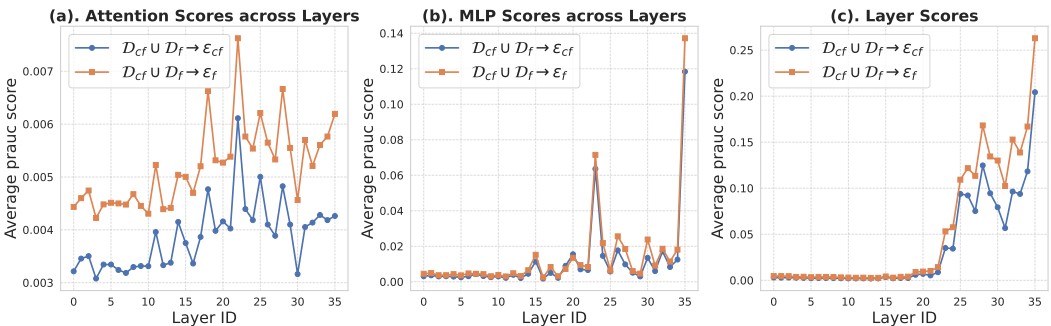

Figure 12: Information intensity probing results (measured by PRAUC scores) for Qwen 3.0 model using the union of original and counterfactual documents $\mathcal{D}_{cf} \cup \mathcal{D}_f$ as context. $\mathcal{D}_{cf} \cup \mathcal{D}_f \rightarrow \mathcal{E}_{cf}$ and $\mathcal{D}_{cf} \cup \mathcal{D}_f \rightarrow \mathcal{E}_f$ represent the average information intensity of counterfactual entities and factual entities respectively.

Table 6: Prompt template for entity-aware citation selection in reading comprehension tasks.

You are an expert in reading comprehension *and* entity-aware citation analysis. You will be given a passage with sentences segmented by identifiers such as <c0>, entity information for each sentence, and a question.

Please read the document, entity information, and question carefully. Use step-by-step reasoning to analyze which sentences or paragraphs need to be cited to answer this question, considering both the semantic content and the entities present in each sentence. Then return a list of citations in the format: `<cite>[citation index only numbers]</cite>`.

**Example:**
[Document Start]
<C0>In the last two decades, several hundred planets have been detected beyond our Solar-system. <C1>Most of these extra-solar planets orbit sun-like stars. <C2>A small number have been detected around stars that are in their late evolutionary state, such as Red Giant Branch (RGB) stars and pulsars. <C3>The phase directly after the RGB stage, the Horizontal Branch (HB), however, is still unexplored; therefore, there is no empirical evidence for whether close-in planets, i.e., those with semi-major axes less than 0.1 AU, survive the giant phase of their host stars.

<C4>Besides its evolutionary stage, a star's chemical composition appears to be a major indicator of its probability for hosting a planet. <C5>Previous studies, e.g.,, showed that main-sequence (MS) stars that host giant planets are metal-rich. <C6>This finding is supported by the large exoplanet search surveys around MS stars reporting a connection between planet frequency and metallicity, and a survey of 160 metal-poor main-sequence stars finding no evidence for Jovian planets.
<C7>Until now, only very few planets have been detected around stars with metallicities as low as [Fe/H]= $-1$, i.e. 10% of the sun's metallicity. <C8>The detection of PSR B1620 b, a Jovian planet orbiting a pulsar in the core of the metal-poor globular cluster M4 ([Fe/H]=$-1.2$), suggests, however, that planets may form around metal-poor stars, although the formation mechanism of this particular planet might be linked to the dense cluster environment.
<C9>We used the Fibre-fed Extended Range Optical Spectrograph (FEROS), a high-resolution spectrograph ($R=48,000$) attached to the 2.2 meter Max-Planck Gesellschaft/European Southern Observatory (MPG/ESO) telescope1, to observe the star. <C10>This star is classified as a red HB (RHB) star (Fig. 1) and its metal content is $[Fe/H]\_\mathrm{mean} = -2.09$, i.e. about 1% that of the Sun. <C11>So far, is not known as a binary system. <C12>Detailed stellar parameters can be found in Supporting Online Material, Section [text1].
<C13>Previous radial velocity (RV) measurements of showed a systematic velocity of about 300 (km s)^{-1} with respect to the Sun, indicating that the star belongs to the stellar halo. <C14>Indeed, the star has been connected to the Helmi stream, a group of stars that share similar orbital parameters that stand apart from those of the bulk of other stars in the solar neighborhood. <C15>The Helmi stream stars move on prograde eccentric orbits (R_peri~7 kpc, R_apo~16 kpc) that reach distances up to \vert z\vert_max~13 kpc above and below the galactic plane. <C16>From that, it has been concluded that these stars were once bound to a satellite galaxy of the Milky Way that was tidally disrupted 6-9 Ga ago.
<C17>, with a mean metallicity estimate of [Fe/H]=-2.1, is far more metal-poor than any previously known exoplanet hosting star (Fig. 3). <C18>For the existing theories of hot giant planet formation, metallicity is an important parameter: in particular, it is fundamental for the core-accretion planet formation model. <C19>It might be that initially, in the planet formation phase, had a higher metallicity, and that during its subsequent evolution, it lost its heavier elements. <C20>For example, during the giant phase, heavy elements could have had been incorporated into dust grains and then separated from the star's atmosphere. <C21>However, given the star's membership to the Helmi stream, in which the most metal-rich sub-dwarfs known so far have [Fe/H]$~-1.5$, we do not expect its initial Fe abundance to exceed this value.
[Document End]

[Entity Information]
C0: ["planets", "Solar-system"]
C1: ["extra-solar planets", "sun-like stars"]
C2: ["Red Giant Branch", "RGB", "pulsars"]
...
C6: ["exoplanet search surveys", "MS stars", "metallicity"]
C7: ["planets", "metallicities", "sun's metallicity"]
C8: ["PSR B1620 b", "Jovian planet", "pulsar", "M4"]
C10: ["red HB", "RHB", "Sun"]
C17: ["metallicity", "exoplanet hosting star"]

[Question]
How does the metallicity of PSR B1620 b, at around 1% of the Sun's,
compare to other known exoplanet host stars?

[Step-by-Step Analysis]
Let me analyze this question step by step:
1.  Identify key entities and concepts in the question:
    - "PSR B1620 b" (specific exoplanet)
    - "metallicity" (chemical composition measure)
    - "1% of the Sun's" (metallicity level)
    - "exoplanet host stars" (comparison group)
2.  Find sentences mentioning PSR B1620 b:
    - C8 mentions "PSR B1620 b" as a Jovian planet with metallicity
[Fe/H]=-1.2
    - Need to check if 1% matches this description
3.  Find sentences about metallicity comparisons:
    - C6 discusses the connection between metallicity and planet
frequency
    - C7 mentions "10% of the sun's metallicity" for [Fe/H]=-1
    - C10 mentions "about 1% that of the Sun" for [Fe/H]=-2.09
    - C17 states the star is "far more metal-poor than any previously
known exoplanet hosting star"
4.  Determine which sentences are needed:
    - C6:  Establishes that metal-rich stars host giant planets
    - C7:  Provides context about very few planets around low-metallicity
stars
    - C10:  Provides the 1% metallicity reference point
    - C17:  Makes the direct comparison to other exoplanet host stars

[Answer Citation List]
<cite>[6,7,10,17]</cite>

**Now get ready to handle the following test case.**

[Document Start]
<context>
[Document End]

[Entity Information]
<entities>

[Question]
<question>

[Step-by-Step Analysis]
Let me analyze this question step by step:
1.  Identify key entities and concepts in the question:
2.  Find relevant sentences based on entities and content:
3.  Determine relationships and connections:
4.  Select the minimum necessary citations:

[Answer Citation List]

# G   Dataset example

Table 7: Example from the HotpotQA dataset

| Field | Content |
|---|---|
| **Query** | Which American conglomerate company is a subsidiary of Trump Entertainment Resorts? |
| **Answer** | Icahn Enterprises L.P. |
| **Target Sentences** | It has been a subsidiary of Icahn Enterprises since 2016. Icahn Enterprises L.P. is an American conglomerate company headquartered at the General Motors Building in New York City, New York. |
| **Evidence Index** | [7, 8] |
| **Dataset** | HotpotQA |
| **Language** | English |
| **Index ID** | 5ac05fb55542997d64295a0b |
| **Context (Partial)** | <C0> TVS Group is an Indian diversified industrial conglomerate with its principal headquarters located in Madurai and presence across the Globe. <C1> Almost all holdings of the group are private. <C2> The largest and most visible subsidiary is TVS Motor Company, the third-largest two-wheeler manufacturers in India. <C3> TVS Group, with group revenue of more than US$6 billion, is an automotive conglomerate company, specialized in manufacturing of two-wheeler, three-wheeler, auto-electricals components, high tensile fasteners, die casting products, dealership business, brakes, wheels, tyres, axles, seating systems, fuel injection components, electronic and electrical components and many more. <C4> Trump Entertainment Resorts, Inc. was a gaming and hospitality company that owned and operated the now shuttered Trump Taj Mahal hotel and casino, as well as the now shuttered Trump Plaza Hotel and Casino and the Trump Marina located in Atlantic City, New Jersey, United States. <C5> Formerly known as Trump Hotels & Casino Resorts, it was founded in 1995 by Donald Trump, now 45th President of the United States, who has not had any formal role in the company since 2011, if not earlier. <C6> The company filed for bankruptcy in 2004, 2009 and 2014. <C7> It has been a subsidiary of Icahn Enterprises since 2016. <C8> Icahn Enterprises L.P. is an American conglomerate company headquartered at the General Motors Building in New York City, New York. <C9> The company has investments in various industries including auto parts, energy, metals, rail cars, casinos, food packaging, real estate and home fashion. <C10> The company is currently controlled by investor Carl Icahn. . . . |

Table 8: Example from the SQuAD dataset

| Field | Content |
|---|---|
| **Query** | Where is there a growing presence of Theravada? |
| **Answer** | the west |
| **Target Sentences** | [Theravāda is primarily practiced today in Sri Lanka, Burma, Laos, Thailand, Cambodia as well as small portions of China, Vietnam, Malaysia and Bangladesh. It has a growing presence in the west.] |
| **Evidence Index** | 1 |
| **Dataset** | SQuAD |
| **Language** | English |
| **Index ID** | 56d291c759d6e414001460be |

| Field | Content |
| --- | --- |
| **Context (Partial)** | <C0> Congo-Brazzaville has had a multi-party political system since the early 1990s, although the system is heavily dominated by President Denis Sassou Nguesso; he has lacked serious competition in the presidential elections held under his rule. Sassou Nguesso is backed by his own Congolese Labour Party (French: Parti Congolais du Travail) as well as a range of smaller parties. <C1> Theravāda is primarily practiced today in Sri Lanka, Burma, Laos, Thailand, Cambodia as well as small portions of China, Vietnam, Malaysia and Bangladesh. It has a growing presence in the west. <C2> Beyoncé announced a hiatus from her music career in January 2010, heeding her mother's advice, "to live life, to be inspired by things again". During the break she and her father parted ways as business partners. Beyoncé's musical break lasted nine months and saw her visit multiple European cities, the Great Wall of China, the Egyptian pyramids, Australia, English music festivals and various museums and ballet performances. <C3> The SI unit of illuminance and luminous emittance, being the luminous power per area, is measured in Lux. It is used in photometry as a measure of the intensity, as perceived by the human eye, of light that hits or passes through a surface. It is analogous to the radiometric unit watts per square metre, but with the power at each wavelength weighted according to the luminosity function, a standardized model of human visual brightness perception. In English, "lux" is used in both singular and plural. <C4> Chopin's life and his relations with George Sand have been fictionalized in numerous films. The 1945 biographical film A Song to Remember earned Cornel Wilde an Academy Award nomination as Best Actor for his portrayal of the composer. Other film treatments have included: La valse de l'adieu (France, 1928) by Henry Roussel, with Pierre Blanchar as Chopin; Impromptu (1991), starring Hugh Grant as Chopin; La note bleue (1991); and Chopin: Desire for Love (2002). <C5> West has additionally appeared and participated in many fundraisers, benefit concerts, and has done community work for Hurricane Katrina relief, the Kanye West Foundation, the Millions More Movement, 100 Black Men of America, a Live Earth concert benefit, World Water Day rally and march, Nike runs, and a MTV special helping young Iraq War veterans who struggle through debt and PTSD a second chance after returning home. 
 . . . |

Table 9: Example from the TriviaQA dataset

| Field | Content |
| --- | --- |
| **Query** | Who, c1819, wrote the poem *Ode to the West Wind*? |
| **Answer** | Percy Bysshe Shelley |
| **Target Sentences** | [1. Ode to the West Wind is an ode, written by Percy Bysshe Shelley in 1819 near Florence, Italy.] |
| **Evidence Index** | 0 |
| **Dataset** | TriviaQA |
| **Language** | English |
| **Index ID** | sfq_22569–Ode_to_the_West_Wind.txt |

Table 10: Prompt Template (Causal Intervention via Zeroing Attention Heads)

**Template**

```
## Document:  {Document_Content}
You are a helpful QA assistant.  Please read the document above and answer the
following question.  Please reason step by step, and put your final answer within
\boxed{}.
## Question:  {Question_Content}
## Options:  A: {A_Content}, B: {B_Content}, C: {C_Content}, D: {D_Content}
```

**Table 9 – continued from previous page**

| Field | Content |
|---|---|
| **Context (Partial)** | <C0> Ode to the West Wind is an ode, written by Percy Bysshe Shelley in 1819 near Florence, Italy. It was published in 1820 in London as part of the collection *Prometheus Unbound*. The poem expresses hope that its words will inspire change and revolution. <C1> The Wang River is a river in northern Thailand. It flows from the Phi Pan Nam Range in Chiang Rai Province to the Ping River in Tak Province. <C2> An empire is an aggregate of nations ruled by an emperor or sovereign. It can be territorial or maritime, often associated with imperialism or colonialism. <C3> "The rum ration (also called tot) was a daily amount of rum given to sailors on Royal Navy ships. It was abolished in 1970 after concerns that regular intakes of alcohol would lead to unsteady hands when working machinery. Tradition: The rum ration, or "tot", consisted of rum at 95.5 proof (54.6% ABV), given out to every sailor at midday. Senior Ratings (Petty Officers and above) received their rum neat, whilst for Junior Ratings it was diluted with two parts of water to make grog. The rum ration was served from one particular barrel, also known as the "Rum Tub" which was ornately decorated and was made of oak, reinforced with brass bands, with brass letters saying, "The Queen, God Bless Her". Not all sailors necessarily drew their rum – each had the option to be marked in the ship's books as "G" (for Grog) or "T" (for Temperate) if there were members of the Temperance Movement. Sailors who opted to be "T" were given three pence (3d) a day instead of the rum ration, although very few sailors took this option. Instead they gave away their ration in exchange for favours. The time when the rum ration was distributed was called "Up Spirits", which was between 11 am and 12 noon. A common cry from the sailors was "Stand fast the Holy Ghost". Each mess had a "Rum Bosun" who would collect the rum from the officer responsible for measuring the right number of tots for each mess. The officers did not get a rum ration. Tot glasses were kept separate from any other glasses. They were washed on the outside, but never inside, in the belief that residue of past tots would stick to the side of the glass and make the tot even stronger. Sailors under 20 were not permitted a rum ration and were marked on the ship's books as "UA" (Under Age). History: The rum ration was originally beer with a daily ration of one gallon. This official allowance continued till after the Napoleonic Wars. When beer was not available, as it would often spoil easily, it could be substituted by a pint of wine or half a pint of spirits depending on what was locally available. In later years, the political influence of the West Indian planters led to rum being given the preference over arrack and other spirits. The half pint of spirits was originally issued neat; it is said that sailors would "prove" its strength by checking that gunpowder doused with rum would still burn (thus verifying that rum was at least 57% ABV). . . . |

Table 11: Prompt Template (Attention Redistribution via Prompt Manipulation)

**Template**

```
Context:  {Context}
Question:  {Question_Content}
Choices:
A. ...
B. ...
C. ...
D. ...
You don't need to analyze, just output your choice.
```

## H    Supplementary Experiments for Competing Hypotheses

We employ three methods to enhance the model's attention to specific information within the prompt, and observe how the model's responses change accordingly. To enhance the model's attention to counterfactual information, we adopt the following three prompt construction strategies:

1. **Duplicate and append**: Duplicate the counterfactual sentences and append them to the original context to reinforce their salience through repetition (see Table 12).

2. **Insert prompt randomly**: Extract all counterfactual sentences, concatenate them, prepend an emphatic instruction, and insert the resulting string at a random position in the original context (see Table 13).

3. **Tag and append**: Extract and concatenate counterfactual sentences, enclose them with `<must pay attention to>` tags for emphasis, and append this segment to the original context (see Table 14).

Table 12: Duplicate and append method

| Component | Content |
| --- | --- |
| Description | Duplicate the counterfactual sentences and append them to the end of the original context to reinforce their salience through repetition. |
| Example | Context: The Øresund Region is a transnational metropolitan area in northern Europe, centered around the Øresund strait and the cities of Copenhagen, Denmark, and Malmö, Sweden. The region is connected by the Øresund Bridge, which runs nearly 8 km from the Swedish coast to the artificial island Peberholm in the middle of the strait. [...] |
| | The bridge connecting Copenhagen and Malmö is 12 km long. |
| | [... Original context continues with additional infrastructure details and historical facts ...] |
| | The bridge connecting Copenhagen and Malmö is 12 km long. |

Table 13: Insert prompt randomly method

| Component | Content |
| --- | --- |
| Description | Extract all counterfactual sentences, concatenate them, prepend an emphatic instruction, and randomly insert the block into the context. |

| Component | Content |
|---|---|
| Example | Context: The Øresund Region is a transnational metropolitan area in northern Europe, centered around the Øresund strait and the cities of Copenhagen, Denmark, and Malmö, Sweden. The region is connected by the Øresund Bridge, which runs nearly 8 km from the Swedish coast to the artificial island Peberholm in the middle of the strait. [...] |
| | Note: The following information is important and should be taken into account. The bridge connecting Copenhagen and Malmö is 12 km long. |
| | [... Original context continues with additional infrastructure details and historical facts ...] |

Table 14: Tag and append method

| Component | Content |
|---|---|
| Description | Extract and concatenate counterfactual sentences, wrap them with `<must pay attention to>` tags, and append this segment to the context. |
| Example | Context: The Øresund Region is a transnational metropolitan area in northern Europe, centered around the Øresund strait and the cities of Copenhagen, Denmark, and Malmö, Sweden. The region is connected by the Øresund Bridge, which runs nearly 8 km from the Swedish coast to the artificial island Peberholm in the middle of the strait. [...] |
| | `<must pay attention to>` The bridge connecting Copenhagen and Malmö is 12 km long. `</must pay attention to>` |

