# OpenReview forum: "Understanding Parametric and Contextual Knowledge Reconciliation within Large Language Models"
_NeurIPS.cc/2025/Conference — NeurIPS 2025 spotlight_

### Official Review · Reviewer_oNGe · 2025-06-09

**Clarity:** 3
**Significance:** 3
**Originality:** 2
**Rating:** 5
**Confidence:** 4

**Summary:**

This paper introduces a novel entity-aware probing framework to investigate how LLMs reconcile potentially conflicting parametric and contextual knowledge in RAG settings. The authors reinterpret transformer computation through residual streams and design a LoRA-based probing method that can detect ad-hoc entity relevance without altering base model parameters. Through extensive experiments across QA benchmarks and counterfactual setups, the authors reveal that contextual and parametric knowledge are routed by distinct attention heads, exhibit asymmetric propagation dynamics, and interact through a mechanism of superpositional reconciliation.

**Questions:**

* Have the authors considered how toxic contextual knowledge that might trigger LLMs' safety mechanisms affects internal knowledge reconciliation? Would similar patterns of attention routing still hold?
* To what extent do the \[START\] and \[END\] tokens introduced for probing influence the internal computation of LLMs? Specifically, how much attention do LLMs allocate to these OOD tokens?

**Ethical Concerns:**

["NO or VERY MINOR ethics concerns only"]

**Final Justification:**

Since the authors addressed most of our concerns, I decided to raise the rating to 5.

**Limitations:**

Yes.

**Paper Formatting Concerns:**

No.

**Quality:**

3

**Strengths And Weaknesses:**

Strengths:

* The proposed Attention Competition Hypothesis and MLP Arbitration Hypothesis provide novel insights for interpreting conflicting knowledge processing. These hypotheses are empirically validated through a carefully designed probing framework that traces entity-specific signals across transformer layers.
* The distinction between attention head specialization and MLP-based accumulation mechanisms provides a refined understanding of LLM internal processes. The discovery that conflicting knowledge can coexist in residual streams is particularly revealing.
* The authors conduct extensive and controlled experiments to validate their hypotheses. Notably, they introduce a key ablation baseline to demonstrate that probe predictions depend on the internal knowledge processing of the base LLM, rather than the capacity of the LoRA adapter.

Weaknesses:

* While the entity-aware probing framework avoids modifying base model weights, it introduces LoRA adapters and injects special tokens (\[START\], \[END\]) into the input. These interventions may perturb the native inference pathways. Consequently, the behavior traced by the probe may deviate subtly from that of the vanilla LLM, raising concerns about the fidelity of the interpretability conclusions.
* A key claim is that conflicting knowledge persists in residual streams and is superpositionally reconciled rather than being suppressed. However, this assertion is not sufficiently substantiated with targeted experiments. For example, the authors do not investigate whether zeroing out high-information-intensity attention heads would alter the final model prediction.

---

> ### Author Rebuttal · Authors · 2025-07-31
>
> **Dear Reviewer oNGe,**
>
> We sincerely thank you for your constructive and insightful review. Your positive acknowledgment of `our novel insights for interpreting conflicting knowledge processing`, `carefully designed probing framework`, and `refined understanding of LLM internal processes` is particularly encouraging and validates our research approach. We address your questions and concerns as follows:
>
> > **W1.** While the entity-aware probing framework avoids modifying base model weights, it introduces LoRA adapters and injects special tokens ([START], [END]) into the input. These interventions may perturb the native inference pathways. Consequently, the behavior traced by the probe may deviate subtly from that of the vanilla LLM, raising concerns about the fidelity of the interpretability conclusions.
>
> The probe test results demonstrate high correlation with the model's actual output behavior. Additional analysis based on the LLaMA 3.1 model is provided in Appendix F of our paper, with results summarized in Table 1 below. The methods "Duplicate and append," "Insert prompt randomly," and "Tag and append" can be viewed as three approaches to enhance the model's attention to counterfactual knowledge in the context (detailed descriptions are available in Appendix F).
>
> **Table 1: Model Response Patterns Under Different Context Conditions**
> | Context Type | Factual Knowledge Response | Counterfactual Knowledge Response | Other |
> |-|-|-|-|
> | $\mathcal{D}\_{cf}$ | 22.7% | 54.6% | 22.7% |
> | $\mathcal{D}\_{cf}$ (Duplicate and append) | 22.8% | 55.6% | 21.6% |
> | $\mathcal{D}\_{cf}$ (Insert prompt randomly) | 22.8% | 59.9% | 17.3% |
> | $\mathcal{D}\_{cf}$ (Tag and append) | 23.2% | 51.3% | 25.5% |
> | $\mathcal{D}\_{f}\cup \mathcal{D}\_{cf}$ | 56.4% | 27.7% | 15.9% |
> | $\mathcal{D}\_{f}\cup \mathcal{D}\_{cf}$ (Duplicate and append) | 40.2% (↓16.2%) | 45.3% (↑17.6%) | 14.5% |
> | $\mathcal{D}\_{f}\cup \mathcal{D}\_{cf}$ (Insert prompt randomly) | 51.1% (↓5.3%) | 40.9% (↑13.2%) | 8.0% |
> | $\mathcal{D}\_{f}\cup \mathcal{D}\_{cf}$ (Tag and append) | 44.9% (↓11.5%) | 41.2% (↑13.6%) | 13.8% |
>
>
> As shown in Table 1, when the context contains only counterfactual documents, 54.6% of responses follow counterfactual knowledge, significantly higher than responses following factual knowledge (22.7%). This phenomenon aligns with Figure 4(c) in our paper, where the information intensity of counterfactual knowledge in the output layer exceeds that of factual knowledge, and the magnitude of information intensity determines the degree of influence that response knowledge has on the output.
>
> When we enhance the model's attention to counterfactual knowledge in the context through three different methods, the model's output does not shift from counterfactual to factual knowledge. This is consistent with our probing conclusion that "knowledge competition does not occur across categories"—that is, enhancing attention to contextual knowledge does not affect the routing of memory heads to parametric knowledge.
>
> When the input contains both factual and counterfactual documents ($\mathcal{D}\_f\cup \mathcal{D}\_{cf}$ setting), compared to the $\mathcal{D}\_f$ setting, the model's output clearly begins to favor factual knowledge (56.4% vs. 27.7%). Why does the model favor factual knowledge when both types appear simultaneously in the context? Our probing conclusion of "superpositional reconciliation" explains this well. Factual knowledge from the context and factual knowledge from parameters form a superposition, making its information intensity exceed that of counterfactual knowledge sourced only from the context. Since information intensity magnitude directly affects output, the model favors factual knowledge.
>
> Subsequently, when we enhance the model's attention to counterfactual knowledge, the proportion of counterfactual responses increases significantly. The results of this operation under $\mathcal{D}\_f\cup \mathcal{D}\_{cf}$ setting are strikingly different from those under $\mathcal{D}\_f$ setting, but this is not surprising. Under $\mathcal{D}\_f\cup \mathcal{D}\_{cf}$ setting, factual knowledge comes from both model parameters and context. When we enhance the model's attention to counterfactual context, the two compete for the attention of context heads, weakening the context heads' attention to factual knowledge. This aligns with our probing conclusion that "knowledge competition occurs within categories."
>
> We understand the reviewers' concerns. Developing better probing techniques to accurately capture the internal knowledge processing mechanisms of models remains an important and open research question that merits continued exploration and improvement.
>
> > **W2.** A key claim is that conflicting knowledge persists in residual streams and is superpositionally reconciled rather than being suppressed. However, this assertion is not sufficiently substantiated with targeted experiments. For example, the authors do not investigate whether zeroing out high-information-intensity attention heads would alter the final model prediction.
>
> Following the reviewer's suggestion, we provide experimental results for zeroing out context heads and memory heads in Table 2. Clearing the top-10 context heads and memory heads with the highest information intensity causally affects the model's output. This experimental finding, together with the correlation analysis provided in our response to W1, sufficiently demonstrates the interpretability of information intensity for model output behavior.
>
> **Table 2:The causal impact of context heads and memory heads on model output behavior**
> | Experimental Setting | Response Consistent with Factual Knowledge | Response Consistent with Counterfactual Knowledge | Other |
> |---------------------|-------------------------------------------|--------------------------------------------------|-------|
> | $\mathcal{D}\_{cf}$ | 22.7% | 54.6% | 22.7% |
> | $\mathcal{D}\_{cf}$ w/o Context Head | 34.9% | 43.6% | 21.5% |
> | $\mathcal{D}\_{cf}$ w/o Memory Head | 9.2% | 65.5% | 25.3% |
>
> The response addresses the reviewer's concern about insufficient experimental substantiation by providing targeted ablation experiments that demonstrate the causal impact of high-information-intensity attention heads on model predictions, thereby supporting the claim about superpositional reconciliation of conflicting knowledge in residual streams.
>
> > **Q1.** Have the authors considered how toxic contextual knowledge that might trigger LLMs' safety mechanisms affects internal knowledge reconciliation? Would similar patterns of attention routing still hold?
>
> Thank you for raising this important point. We hypothesize that toxic contextual knowledge may exhibit distinct internal processing patterns. RLHF alignment training could potentially suppress contextual heads' attention to such harmful knowledge. This represents a critical direction for future research with significant implications for RAG system robustness and alignment. While our current findings are limited to factual knowledge conflicts, they provide a foundational methodology that can be extended to investigate safety-knowledge interactions—an important area we hope to explore in subsequent work.
>
>
> > **Q2.** To what extent do the [START] and [END] tokens introduced for probing influence the internal computation of LLMs? Specifically, how much attention do LLMs allocate to these OOD tokens?
>
> Based on the responses to W1 and W2, we can see that the probe results align well with the model's actual output behavior. We believe the influence of [START] and [END] tokens, as well as LoRA updates, on LLM working modes can be understood as follows. Under normal operating conditions, LLMs respond to user instructions by continuously predicting the next token. At each time step t, their computation is biased toward the token currently being predicted. Obviously, not every target token at each step is knowledge-intensive. We believe that [START] and [END] tokens may serve as indicators for the model to simulate the computations executed when predicting tokens corresponding to the target entity.

---

> > ### Comment · Reviewer_oNGe · 2025-08-04
> >
> > Since the authors addressed most of our concerns, I decided to raise the rating to 5.

---

> > > ### Author Response · Authors · 2025-08-04
> > > **Official Comment by the Authors**
> > >
> > > We sincerely appreciate your insightful comments once again. We're glad the additional experiments and clarifications addressed your questions. Thank you for your continued support!

---

### Official Review · Reviewer_6M6N · 2025-06-23

**Clarity:** 3
**Significance:** 4
**Originality:** 3
**Rating:** 5
**Confidence:** 4

**Summary:**

This paper provides a rigorous analysis and mechanistic understanding of how LLMs reconcile conflicting parametric knowledge (internalized in the model parameters during pretraining) and contextual knowledge (present as in context tokens) in retrieval-augmented generation/RAG settings.

Specifically the authors propose a novel entity based probing technique which leverages specially appended tokens and a finetuned LoRA adapter to track entity-level relevance across transformer layers, without fine-tuning the base model.
Using this framework they probe the information of a specified entity at 3 computation points within each transformer layer: MLP update, attention update and layer output.

They find several interesting patterns, consistent across different models, namely that
1. parametric and contextual knowledge utilize distinct pathways revealing attention head specialization;
2. contextual knowledge emerges more abruptly whereas parametric knowledge accumulates more gradually through MLP layers;
3. super-positional reconciliation shows conflicting knowledge persists in residual streams with similar information from multiple sources reinforcing each other over contradicting information e.g. parametric + factual document versus counterfactual document information.

**Questions:**

Could authors more clearly explain around why the entity triple being appended does not affect the output? For example even with a randomly initialized LLM the mean F1 is around 20 (one would expect zero for complete independence)

**Ethical Concerns:**

["NO or VERY MINOR ethics concerns only"]

**Final Justification:**

My original score was already at a 5. I had a few small questions/clarifications/concerns which authors' have addressed, no change to my original rating.

**Limitations:**

Yes

**Paper Formatting Concerns:**

Line 48; phrase "Our analysis reveals three key insights" repeated twice
Line 68: "rewrite" should be "rewriting"

**Quality:**

4

**Strengths And Weaknesses:**

Strengths:
1. The approach described as far as I know is quite novel.
2. The information conflict between parametric and contextual knowledge and controlling/understanding it is a topic of high importance for academic and industrial research.
3. The experimental approach is very rigorous and well through out.
4. Experiments are robust with comprehensive and well-motivated ablations and conclusions.
5. Reveals extremely interesting and insightful patterns around where and how parametric and contextual information is aggregated in the LLM and how knowledge conflicts are handled.

Weaknesses:
1. I am still less convinced on the triple being appended to the model input not affecting the outputs and conclusions at all. The authors attempt to isolate the impact of the probe by comparing performance on a randomly initialized model (where performance collapses) however it seems counterintuitive that this would have no impact and there are not better ways e.g. probing around the token the entity organically appears in the context (at least for the cases where it does)

---

> ### Author Rebuttal · Authors · 2025-07-31
>
> **Dear Reviewer 6M6N,**
>
> We are grateful for your thorough review and positive assessment. Your recognition of `the novelty of our approach`, `the significant value of our research topic`, and `the rigor of our experiment designs` along with `the depth of our conclusions` motivates us and confirms the value of our research contributions. We address your questions and concerns as follows:
>
> > **W1.** I am still less convinced on the triple being appended to the model input not affecting the outputs and conclusions at all. The authors attempt to isolate the impact of the probe by comparing performance on a randomly initialized model (where performance collapses) however it seems counterintuitive that this would have no impact and there are not better ways e.g. probing around the token the entity organically appears in the context (at least for the cases where it does)
> >
> > **Q1.** Could authors more clearly explain around why the entity triple being appended does not affect the output? For example even with a randomly initialized LLM the mean F1 is around 20 (one would expect zero for complete independence)
>
> We break down W1&Q1 into several sub-questions and address them one by one.
>
> **(1) Why does a randomly initialized LLM achieve an average F1 score around 20? Is this reasonable?**
>
> An F1 score around 20 represents the result of random guessing. Let us illustrate this with a concrete example: suppose your attribution evaluation dataset consists of samples where each contains a query and 100 candidate sentences, with 10 target sentences containing information relevant to the query. Your task is to identify these 10 target sentences from the 100 candidates.
>
> Now consider a "poor" model that completely ignores the semantic content of candidate sentences and considers all of them relevant to the query—essentially predicting all 100 candidates as target sentences. Let's calculate the F1 score in this scenario:
>
> Precision = $\frac{\text{Number of correctly identified targets}}{\text{Number of predicted targets}} = \frac{10}{100} = 0.1$
>
> Recall = $\frac{\text{Number of correctly identified targets}}{\text{Total number of actual targets}} = \frac{10}{10} = 1.0$
>
> F1 = $2 × \frac{\text{precision × recall}}{\text{(precision + recall)}} = 0.182$
>
> Clearly, an F1 score of 0.182 does not indicate that this "poor" model possesses any attribution capability.
>
>
> **(2) Since probes require specific input formats, are the probing results consistent with the model's actual behavioral patterns?**
>
> The probe test results demonstrate high correlation with the model's actual output behavior. Additional analysis based on the LLaMA 3.1 model is provided in Appendix F of our paper, with results summarized in Table 1 below. The methods "Duplicate and append," "Insert prompt randomly," and "Tag and append" can be viewed as three approaches to enhance the model's attention to counterfactual knowledge in the context (detailed descriptions are available in Appendix F).
>
> **Table 1: Model Response Patterns Under Different Context Conditions**
> | Context Type | Factual Knowledge Response | Counterfactual Knowledge Response | Other |
> |-|-|-|-|
> | $\mathcal{D}\_{cf}$ | 22.7% | 54.6% | 22.7% |
> | $\mathcal{D}\_{cf}$ (Duplicate and append) | 22.8% | 55.6% | 21.6% |
> | $\mathcal{D}\_{cf}$ (Insert prompt randomly) | 22.8% | 59.9% | 17.3% |
> | $\mathcal{D}\_{cf}$ (Tag and append) | 23.2% | 51.3% | 25.5% |
> | $\mathcal{D}\_{f}\cup \mathcal{D}\_{cf}$ | 56.4% | 27.7% | 15.9% |
> | $\mathcal{D}\_{f}\cup \mathcal{D}\_{cf}$ (Duplicate and append) | 40.2% (↓16.2%) | 45.3% (↑17.6%) | 14.5% |
> | $\mathcal{D}\_{f}\cup \mathcal{D}\_{cf}$ (Insert prompt randomly) | 51.1% (↓5.3%) | 40.9% (↑13.2%) | 8.0% |
> | $\mathcal{D}\_{f}\cup \mathcal{D}\_{cf}$ (Tag and append) | 44.9% (↓11.5%) | 41.2% (↑13.6%) | 13.8% |
>
>
> As shown in Table 1, when the context contains only counterfactual documents, 54.6% of responses follow counterfactual knowledge, significantly higher than responses following factual knowledge (22.7%). This phenomenon aligns with Figure 4(c) in our paper, where the information intensity of counterfactual knowledge in the output layer exceeds that of factual knowledge, and the magnitude of information intensity determines the degree of influence that response knowledge has on the output.
>
> When we enhance the model's attention to counterfactual knowledge in the context through three different methods, the model's output does not shift from counterfactual to factual knowledge. This is consistent with our probing conclusion that "knowledge competition does not occur across categories"—that is, enhancing attention to contextual knowledge does not affect the routing of memory heads to parametric knowledge.
>
> When the input contains both factual and counterfactual documents ($\mathcal{D}\_f\cup \mathcal{D}\_{cf}$ setting), compared to the $\mathcal{D}\_f$ setting, the model's output clearly begins to favor factual knowledge (56.4% vs. 27.7%). Why does the model favor factual knowledge when both types appear simultaneously in the context? Our probing conclusion of "superpositional reconciliation" explains this well. Factual knowledge from the context and factual knowledge from parameters form a superposition, making its information intensity exceed that of counterfactual knowledge sourced only from the context. Since information intensity magnitude directly affects output, the model favors factual knowledge.
>
> Subsequently, when we enhance the model's attention to counterfactual knowledge, the proportion of counterfactual responses increases significantly. The results of this operation under $\mathcal{D}\_f\cup \mathcal{D}\_{cf}$ setting are strikingly different from those under $\mathcal{D}\_f$ setting, but this is not surprising. Under $\mathcal{D}\_f\cup \mathcal{D}\_{cf}$ setting, factual knowledge comes from both model parameters and context. When we enhance the model's attention to counterfactual context, the two compete for the attention of context heads, weakening the context heads' attention to factual knowledge. This aligns with our probing conclusion that "knowledge competition occurs within categories."
>
> **(3) Are there better methods? For example, probing around context tokens where entities naturally occur?**
>
> We appreciate the reviewer's suggestion to explore more natural probing methods. Indeed, developing better probing techniques to accurately capture models' internal knowledge processing mechanisms remains an important and open research question worthy of continued exploration and improvement. However, the reviewer's suggested approach of "probing at positions where entities naturally occur in context" faces a fundamental limitation, which is precisely the core problem our entity-aware probing framework aims to address: it cannot effectively probe parametric knowledge.
>
> For example, when an external document claims "Trump is the US President," the model's parametric knowledge "Biden" simply does not appear in the input context. We would therefore be unable to probe the model's internal processing and propagation of the parametric knowledge "Biden." Our method enables dynamic tracking of any relevant entity (regardless of whether it appears in the context) by explicitly marking target entities, which is a necessary condition for understanding LLMs' knowledge reconciliation mechanisms.

---

> ### Comment · Reviewer_6M6N · 2025-08-04
>
> Thanks to authors for taking time to address/answer some of my concerns. Here are my follow ups
>
> Q1: The example of the .20 F1 baseline seems a bit unconvincing, let me explain why. Authors assume a bad model would predict all positives, thereby coming up with a recall of 1.0. However, using another definition of a bad model could be it predicts only 1 TP in which case precision would be 0.01 and recall would be 0.1; therefore F1 would be 0.01.
>
> Q2: Authors in Q2, do not seem to be answering the essence of my question. As in my question was about whether the probing actually is a window into the model's internal behavior patterns or instead reflects byproduct of the prompting setup and it seems they are talking more about information intensities which they analyzed in the results.
>
> Q3: Authors adequately address why my very quick and dirty idea is not applicable and I agree completely. However, my point was not necessarily to follow that approach (it was meant as a mere simplistic example), but rather to ask whether there are alternative probing strategies that avoid modifying self‑attention through appended triples. So let me better re-phrase: is this strategy the "only" way to probe the model which the authors thought about or has been done earlier in the literature? Did they consider any other approaches, or come across others in previous literature? If so, then why did they choose this approach?

---

> > ### Author Response · Authors · 2025-08-04
> > **Response to Reviewer Comments [1/2]**
> >
> > **Dear Reviewer 6M6N,**
> >
> > Thank you for taking the time to engage with our work. We appreciate your careful consideration of our methodology and results. Below, we address each of your concerns in detail:
> >
> > > **Q1**: The example of the .20 F1 baseline seems a bit unconvincing, let me explain why. Authors assume a bad model would predict all positives, thereby coming up with a recall of 1.0. However, using another definition of a bad model could be it predicts only 1 TP in which case precision would be 0.01 and recall would be 0.1; therefore F1 would be 0.01.
> >
> > We completely agree that there exist poor models that would achieve F1 scores close to 0. We want to clarify that the .20 F1 example is not a completely fabricated special case, but rather a demonstrative example that "extremizes" the output behavior of the "Probing with randomized LLM" baseline for easier understanding.
> >
> > In this response, we directly analyze the output behavior of the "Probing with randomized LLM" baseline, hoping this will help address your concerns.
> >
> > Table 1 shows the attribution experimental setup and the probing accuracy of this baseline (using LLaMA3.1 as an example; the other two models show similar patterns). As can be seen, the baseline's probing precision is very close to random selection from candidate documents/sentences. However, the recall is quite high, meaning the model makes a large number of positive predictions with near-random precision, recalling most positive sentences/paragraphs across the three datasets.
> >
> > Overall, we believe such a baseline lacks valuable attribution capability.
> >
> >
> > **Table 1: "Probing with randomized LLM" baseline's positive prediction precision is almost at random guessing level.**
> > | Dataset | Attribution Granularity | Avg # Candidate Docs/Sentences | Avg # Positive Docs/Sentences | Positive-Candidate Ratio | Probing Precision |Probing Recall |
> > | - | - | - | -| -| - | - |
> > | TriviaQA     | Paragraph-level      | 5     |1     |20.0%     |19.9%     |61.7%     |
> > | SQuAD     | Paragraph-level     | 10     |1     |10.0%     |10.0%     |83.4%     |
> > | HotpotQA     | Sentence-level     | 40.8     |2.4     |5.9%     |7.0%     |69.5%     |

---

> > ### Author Response · Authors · 2025-08-04
> > **Response to Reviewer Comments [2/2]**
> >
> > > **Q2**: Authors in Q2, do not seem to be answering the essence of my question. As in my question was about whether the probing actually is a window into the model's internal behavior patterns or instead reflects byproduct of the prompting setup and it seems they are talking more about information intensities which they analyzed in the results.
> > >
> > > **Q3**: Authors adequately address why my very quick and dirty idea is not applicable and I agree completely. However, my point was not necessarily to follow that approach (it was meant as a mere simplistic example), but rather to ask whether there are alternative probing strategies that avoid modifying self‑attention through appended triples. So let me better re-phrase: is this strategy the "only" way to probe the model which the authors thought about or has been done earlier in the literature? Did they consider any other approaches, or come across others in previous literature? If so, then why did they choose this approach?
> >
> > The commonly adopted method in the literatures involves case-by-case design of specific prompts [1,2], observing the model's behavior when generating conflicting entity tokens. For example:
> >
> > > **Document**: The capital of France is London.
> > >
> > > **Q**: Where is the capital of France?
> > >
> > > **A**: The capital of France is ____
> >
> > In this case, the candidate entity tokens include "London" and "Paris." The "next token prediction head" is used to decode the residual stream at the "____" position to analyze how entity tokens are formed within the residual stream.
> >
> > Here, the answer prefix "The capital of France is" serves to elicit the model's conflict reconciliation behavior. In standard generation, not every token prediction involves knowledge-intensive reasoning or conflict resolution. Many tokens (e.g., articles, conjunctions, formatting) are generated through relatively shallow pattern matching.
> >
> > While this approach avoids appending additional triples to explicitly specify the probe targets, it has limitations: (1) it restricts target entities to single-token entities, and (2) case-by-case prompt design hinders scalable testing, with results potentially being highly sensitive to prompting [3].
> >
> > The LoRA updates and the entity triples appended after the document and question in our work can essentially be viewed as a trainable soft prompt. It serves a similar function to "The capital of France is" in eliciting the model's conflict reconciliation behavior. Simultaneously, it avoids case-by-case prompt engineering and eliminates the restriction of limiting target entities to single-token entities. This is the rationale behind our choice of this approach.
> >
> > However, we must acknowledge that this approach increases the risk of altering the model's internal behavioral patterns. We believe that constructing more flexible probing methods with minimal intervention in model behavioral patterns represents an important open problem worthy of continued exploration.
> >
> >
> > [1] Locating and Editing Factual Associations in GPT
> >
> > [2] Cutting Off the Head Ends the Conflict: A Mechanism for Interpreting and Mitigating Knowledge Conflicts in Language Models
> >
> > [3] PTP: Boosting Stability and Performance of Prompt Tuning with Perturbation-Based Regularizer.
> >
> >
> > We hope these clarifications address your concerns. If you have any additional concerns, we would be more than happy to engage in further discussion. Thank you again for your valuable feedback.
> >
> > Best regards,
> >
> > The Authors

---

> ### Comment · Reviewer_6M6N · 2025-08-05
>
> Authors' responses addressed my concerns, no change to my original ratings

---

> > ### Author Response · Authors · 2025-08-05
> > **Official Comment by the Authors**
> >
> > We deeply appreciate your thoughtful feedback. We're pleased that our additional analyses have satisfactorily addressed your concerns. Thank you for your valuable time and expertise!

---

### Official Review · Reviewer_KnAk · 2025-07-02

**Clarity:** 3
**Significance:** 2
**Originality:** 3
**Rating:** 5
**Confidence:** 3

**Summary:**

In RAG,  LLMs are provided with some information in their input context. This information may conflict with the LLM’s internal knowledge. This paper designs a probing method to understand how the contextual and parametric knowledge propagates through the layers and how the conflicts are resolved. What is the role of attention and  MLP layers? The probe consists of a fine-tuned classification head to identify if an entity is relevant to answer the given question. Input to the classifier is the attention, the MLP, or the entire layer residual embedding. If the entity is classified as relevant, then the corresponding source from which the input is taken (attention head, MLP, or the entire layer) is considered relevant for the entity. With counterfactual information in the context, they run the probe for factual and counterfactual entities that are assumed to represent the parametric and contextual knowledge, respectively. The sources (attention, MLP or entire layer) that identify factual entities as relevant are considered important for parametric knowledge, and sources that identify counter-factual entities as relevant are considered important for contextual knowledge.

**Questions:**

1. “Here, contextual knowledge types (factual vs. counterfactual) compete for attention within contextual heads:” Line 265: Once you identify the contextual and memory heads in (i) from fig 3(a) and 3(b), do they remain fixed for (iii)?
For section 5.2 (iii), only the top 10 contextual and memory heads are important. Maybe a better representation would be a bar chart with 10 bars for $E_f$ and 10 bars for $ E_{cf}$; X axis representing top-10 contextual head ids (in descending order); Y axis representing information intensity. That will give a direct comparison between the contextual information intensity gap between $E_f$ and $E_{cf}$. You can repeat the same for memory heads.

2. “However, the presence of factual knowledge in the context has minimal impact on memory heads” line 267: In the case of $E_f$, memory heads and contextual heads are corroborating with each other. Is there any analysis that demonstrates this?

3. “the average information intensity of top-10 memory heads slightly increases” line 269: Are you identifying top-10 memory heads again, or are they the same as identified in section 5.2 (i).

4. The range of prauc scores in Fig. 4a and 5a is not the same. Does that impact your analysis?

5. In Fig. 4a and 5a, how are you aggregating the information intensity across heads?

6. “Contextual knowledge (E_cf ), in contrast, is immediately detectable in early attention updates,” p. 8, line 284:  Isn’t the range on the y-axis for Fig. 4(a) too low to make this comment?

7. Line 303 to 305:  Could you please elaborate on this?

8. “Asymmetric propagation mechanisms reveal contextual knowledge emerges abruptly via attention while parametric knowledge accumulates gradually through MLP layers;” Line 338: Which observation supports the abrupt emergence of contextual knowledge via attention? Gradual accumulation via MLP layers seems to be common to both parametric and contextual knowledge.

**Ethical Concerns:**

["NO or VERY MINOR ethics concerns only"]

**Final Justification:**

My concerns were with the experimental setup, and they have been addressed by the authors during the rebuttal.
Hence, I am raising my score to accept.

**Limitations:**

yes

**Paper Formatting Concerns:**

No formatting concerns.

**Quality:**

2

**Strengths And Weaknesses:**

**Strengths**

1.    The paper aims to understand a pertinent issue that often arises when LLMs are used with dynamic external information. It is important to understand how and when the LLMs rely on contextual information and when they fall back on their parametric knowledge. It is also important from the perspective of designing guardrails for LLMs, as the input context may prompt the model not to follow ethics or may contain profanity, and instructions to override LLM’s internal alignment. Understanding how contextual and parametric knowledge reconcile can help us design better guardrails as well.

2.   The design of counterfactual information is neat and clever. Creation of fictional entities ensures that the contextual information is not in the parametric knowledge.

3.   Design of an experiment to validate the probe using the attribution task is also helpful.


**Weaknesses**

The most crucial part of an analysis paper is the experimental design, and I think there are some issues with it.



**Probe design:**

1.  During the training of the probe, you are not using the counterfactual data. And you assume that the factual entities represent the parametric knowledge (line 219-220).  So are they representing the parametric knowledge or the contextual knowledge?

2. When training the classification head, do you compute loss using the MLP residuals from all the layers and attention residuals from all the heads of all the layers? If not, then can you rely on its output when it is used to quantify information intensity in the intermediate residual streams?

3. In section 4, eqn 4, which residual stream are you using to compute $P_\theta$?

4. You are training a LoRA for the embedding layer. Why not simply learn embeddings of the two new special tokens? Why impact all other embeddings via LoRA?

5. In Probing-based attribution (section 4), you need to first identify the entities in the sentence.  I.e, you have additional information about the entities in the document. If those are extracted using a stronger model like GPT-4o, then that gives probing an unfair advantage.
Maybe for a fair comparison, you could provide the extracted entities as part of the input to the baseline prompting method, and also design a prompt with CoT style one-shot demonstration (the way you do in Table 2 for data generation).

**Using the probe for understanding parametric and contextual knowledge (section 5)**

6. The probe is only identifying if an input entity is relevant to answer the given question or not. I am not sure if its usage to identify processing units (attention heads and MLP layers) for counterfactual (contextual) or parametric knowledge is reasonable or not.

7.  Line 219: “we assume its factual descriptions align with LLMs’ parametric knowledge.”: It is a very crucial assumption, and the entire analysis rests on it. Unfortunately, it may not always be true. If it were always true, then LLMs would get close to 100% performance on HotpotQA even when prompted without any documents, as the required knowledge is in their parameters.
For this analysis, maybe you should select those questions where the LLM gives the correct answer out of the box without providing any external document in the context. This will ensure that documents are truly aligned with LLM's internal parametric knowledge.


**Typos:**

1. “Our analysis reveals three key insights: Our analysis reveals three key insights:” p. 2

2. “individual attention head outputs $∆h_{l}^{attn}(t)$.” p. 3, line 97 - individual attention head output should have $(k)$ as superscript.

3. “Probing-based attribution: We addresses the..” p. 5, line 186

4. “we probing all mentioned entities” p. 5, line 187.

5. “$E − (E_f ∪ E_f )$” : p. 6, line 230: it should be $E_{f} ∪  E_{cf}$

6. “However, how models handle such conflicts remains an area for further research. research."” : p. 9, line 325: there is a closing quote without any opening quote.

7. “The outputs of N attention heads are summed to form the attention sublayer output ...” p. 3, line 91:  $i$  should go from 1 to N and not $k$ to N. The superscript on RHS should be $(i)$ and not $(k)$.

8. “We follow the approach of Ye et al. [41] to resolve the limitations of traditional probing outlined in Section 2.2, adapting the probe fθ for entity relevance probing” p. 4, line 128 to 130 -  Looks like an incorrect citation-- did you mean to cite Physics of Language Models (PoLM) -3.1? You have cited PoLM-2.1.

---

> ### Author Rebuttal · Authors · 2025-07-31
>
> **Dear Reviewer KnAK**,
>
> Thank you very much for taking the time to provide such valuable and detailed feedback on our manuscript. We are pleased that you found merit in `the important value of our research on conflict reconciliation mechanisms`, `our neat and clever counterfactual data synthesis`, and `helpful probing validation experiments`. We address your questions and concerns as follows:
>
> > **W1**. Questions regarding the probe training data and the relationship between entities and their corresponding knowledge types.
>
> **The probe training does utilize counterfactual data**. We generate the probe's training data based on the prompt template provided in Table 2 of the paper. The prompt includes both "Analysis" and "Counterfactual Data Synthesis" components. The latter generates counterfactual versions of original documents and identifies corresponding counterfactual entities. For the three training data types in the probe, **entities map to their knowledge types as follows**:
>
> **Table 1: Probe Training Dataset Categories**
> | Document Type | Positive Entities | Knowledge Type Represented |
> |-|-|-|
> | Factual document $\mathcal{D}_f$ | Factual entities $\mathcal{E}_f$ | Superposition of parametric and contextual knowledge |
> | Counterfactual document $\mathcal{D}_{cf}$ | Factual entities $\mathcal{E}_f$ | Parametric knowledge |
> | Counterfactual document $\mathcal{D}_{cf}$ | Counterfactual entities $\mathcal{E}_{cf}$ | Contextual knowledge |
>
> We commit to improving the clarity of section3.3 to avoid misunderstandings.
>
>
> > **W2&W6**. Concerns about the probe training inputs and whether probes can measure the information strength in residual streams.
>
> Probes are trained on output layer representations, then used to measure information strength in residual streams. This approach is known as ``early exit`` [1-3], which has proven to be effective even without special training processes [4], as the residual connections [5] in transformer layers make the hidden representations gradually evolve without abrupt changes.
>
> [1] Fast inference via early exiting from deep neural networks.
>
> [2] Depth-adaptive transformer.
>
> [3] Confident adaptive language modeling.
>
> [4] Bert's output layer recognizes all hidden layers? some intriguing phenomena and a simple way to boost bert.
>
> [5] Deep residual learning for image recognition.
>
>
> >**W3**. In section 4, eqn 4, which residual stream are you using to compute ?
>
> We compute using the [END] token's output layer representation.
>
> >**W4**. Why use LoRA to implement probes?
>
> As described in line 128 of the paper, we fully follow the approach in [1] for probing entity information, including introducing two new special tokens and a LoRA update. According to the description in Section 4.1 of [1], the "small" LoRA update is designed to help LLMs adapt to changes in input format. We hypothesize that completely freezing LLM parameters and relying solely on two new token embeddings would be insufficient to achieve this adaptation.
>
> [1] Physics of language models: Part 2.1, grade school math and the hidden reasoning process.
>
> >**W5**. Fairness concerns in attribution experiments.
>
> Following the your suggestion, we enhanced the baseline prompting methods with the following conditions to ensure fair comparison:
> - **+entity**: Providing GPT-4o extracted entity information as input.
> - **+CoT**: Incorporating carefully designed CoT-style one-shot demonstrations.
>
> **Table 2: Attribution Results with Fair Comparison Conditions vs. Probing Methods**
> | Model | Average |
> |-|-|
> | Qwen2.5-7B+entity+CoT | 75.7 |
> | Qwen2.5-7B-Probing | 73.0 |
> | Qwen3-8B+entity+CoT | 81.6 |
> | Qwen3-8B-Probing | 70.8 |
> | LLaMA3.1-8B+entity+CoT | 63.1 |
> | LLaMA3.1-8B-Probing | 68.4 |
>
> **Highlight our motivation**: It is important to clarify that our probing method is not intended to compete with prompting methods in terms of performance, but rather to provide a complementary analytical tool for understanding the internal attribution mechanisms of language models.
>
> **Validation of Intrinsic Model Capabilities**: The results demonstrate that under sufficient conditions (+entity+CoT), instruction-following models can indeed "articulate" accurate attributions. This confirms that models intrinsically possess the ability to perceive entity relevance; they simply require appropriate methods to elicit this latent knowledge.
>
>
> >**W7**. Concerns regarding alignment between LLM parametric knowledge and HotpotQA contextual knowledge, and suggestions for testing on filtered subsets.
>
> Following the reviewer's suggestion, we filtered the evaluation data to include only questions that LLMs answer correctly without any document prompts. Our main findings remain consistent on these subsets, including distinct routing pathways, within-type competition, and superpositional reconciliation. The primary difference is that factual entities show higher average information intensity compared to the full dataset. Due to rebuttal length constraints, we will include the complete results in the revised paper.
>
>
> >**Q1&Q3**. Do the context heads and memory heads identified in Figure 3(a) and (b) remain consistent in part (iii) of section 5.2? A bar chart would be a better presentation format.
>
> The context heads and memory heads identified in part (i) from Figure 3(a) and (b) remain consistent in part (iii). There is only one minor exception: the 11th context head in layer 14 falls out of the top 10 in terms of information intensity in subplot (d) "$\mathcal{D}_{cf}\cup \mathcal{D}_f \rightarrow \mathcal{E}_f$". Due to rebuttal policy constraints, we cannot provide graphical results, so we are using tabular data as a substitute. The context heads and memory heads are ranked according to their information intensity from high to low as shown in Figure 3(a) and (b), respectively. We will supplement the results with bar charts in the revised version of the paper.
>
> **Table 3: Distribution and Information Intensity Values of Context Heads**
> |Layer ID|Head ID|Fig. 3(a) $\mathcal{D}_{cf} \rightarrow \mathcal{E}\_{cf} $|Fig. 3(c) $\mathcal{D}_{cf}\cup \mathcal{D}_f \rightarrow \mathcal{E}\_{cf}$|Fig. 3(d) $\mathcal{D}_{cf}\cup \mathcal{D}_f\rightarrow\mathcal{E}_f$|
> |-|-|-|-|-|
> |16|4|0.119|0.086|0.079|
> |15|7|0.119|0.042|0.061|
> |19|18|0.115|0.039|0.045|
> |12|22|0.092|0.052|0.079|
> |22|4|0.089|0.041|0.037|
> |20|15|0.086|0.048|0.047|
> |14|11|0.069|0.028|0.019|
> |24|28|0.066|0.030|0.040|
> |15|0|0.051|0.030|0.030|
> |20|7|0.048|0.029|0.034|
>
> **Table 4: Distribution and Information Intensity Values of Memory Heads**
> |Layer ID|Head ID|Fig. 3(b) $\mathcal{D}\_{cf}\rightarrow\mathcal{E}\_{f}$|Fig. 3(d) $\mathcal{D}\_{cf}\cup \mathcal{D}\_f\rightarrow\mathcal{E}\_f$|
> |-|-|-|-|
> |27|14|0.027|0.028|
> |31|26|0.022|0.010|
> |12|22|0.018|0.079|
> |26|23|0.018|0.011|
> |16|16|0.018|0.015|
> |18|9|0.017|0.019|
> |20|23|0.016|0.018|
> |29|25|0.016|0.019|
> |13|25|0.015|0.026|
> |30|23|0.015|0.019|
>
> >**Q2**. Impact of Factual Knowledge in Context on Memory Heads
>
> Based on Table 4's analysis of 10 memory heads, we address this concern systematically. Among these heads, one overlaps with context heads (Layer 12, Head 22, as mentioned in Line 251).
> When context shifts from $\mathcal{D}_{cf}$ to $\mathcal{D}\_{cf}\cup\mathcal{D}\_{f}$, factual knowledge exists both as contextual and parametric knowledge. The overlapping head (Layer 12, Head 22) shows a 4x+ increase in information intensity (0.018 → 0.079), as it captures both parametric knowledge and is highly sensitive to newly injected factual knowledge in context.
> In contrast, the remaining 9 pure memory heads show much smaller magnitude changes with inconsistent directions (some increase, some decrease). Therefore, we conclude that factual knowledge in context has minimal impact on pure memory heads, at least on average.
>
> >**Q4**. The range of prauc scores in Fig. 4a and 5a is not the same.
>
> The ranges are indeed different, which is an inevitable result of contextual knowledge competition. In Figure 4a, the counterfactual entity monopolizes all attention from the contextual heads. In Figure 5a, factual documents are introduced into the context, where factual entities draw away some of the attention from contextual heads. Therefore, the range of PRAUC scores becomes smaller. This can also be observed in Table 3 from response to Q1&Q3. After introducing $\mathcal{D}_{f}$, the attention from 10 context heads was diverted to factual entities, reducing the average information strength of counterfactual entities from 0.0854 to 0.0425.
>
> >**Q5** How are you aggregating the information intensity across heads?
>
> Measure the aggregated information intensity by probing the output representation of the attention layer (described in line 92 of the paper).
>
> > **Q6&Q8.**: Concerns regarding the formation mechanisms and information intensity of contextual vs. parametric knowledge
>
> In Fig. 4(a), the information intensity of contextual knowledge in early attention updates is already 4-6 times higher than parametric knowledge, representing a substantial gap. This indicates that contextual knowledge emerges in early attention routing, while the complete semantics of parametric knowledge do not yet appear. We hypothesize that parametric knowledge formation requires layer-by-layer abstraction of knowledge fragments until complete semantic units are formed near the output layers.
>
> It's worth noting that residual connections in modern neural networks enable representations to gradually evolve without abrupt changes, making it difficult to observe very large information intensity values.
>
> > **Q7.** Line 303 to 305: Could you please elaborate on this?
>
> This indicates that attention competition occurs only within the same type of knowledge (refer to the analysis of Q4), rather than across different categories (refer to the analysis of Q2).

---

> > ### Comment · Reviewer_KnAk · 2025-08-03
> >
> > Dear authors
> >
> > Thank you for your detailed response. I appreciate the additional experiments to address W5 and W7.
> >
> > After looking at Table 2 above, I looked at Table 1 again and observed that the designed probe achieves an F1 score of around 50 on HotpotQA, whereas on Squad, you obtained an F1 score of around 90. I wonder why you chose HotpotQA to construct your experiments? Given low performance on HotpotQA, can we rely on the PRAUC obtained by the trained classifier? Would it not be better to conduct knowledge reconciliation experiments with Squad instead?
> >
> >
> > Regards,

---

> > > ### Author Response · Authors · 2025-08-03
> > > **Response to Reviewer Comments**
> > >
> > > **Dear Reviewer KnAK**,
> > >
> > > We are pleased that our previous response successfully addressed your concerns regarding "the fairness of comparisons in attribution experiments" and "the alignment between LLMs' parametric knowledge and contextual knowledge in HotpotQA."
> > >
> > > Regarding your new question about our choice of HotpotQA for the experiments, we believe this is an important point that deserves further clarification.
> > >
> > > **Why HotpotQA is a better choice for our experiments**:
> > > Using HotpotQA to construct our experiments is indeed a better choice because it enables us to generate more accurate training and evaluation data. Specifically, the probe's identification target is entities relevant to the question, and subsequent reconciliation analysis revolves around these target entities. However, these target entities are missing in QA datasets, which presents the greatest obstacle in constructing probe training and reconciliation evaluation data. For TriviaQA and SQuAD datasets, automatically identifying target entities from entire documents may introduce significant noise. Fortunately, HotpotQA provides sentence-level answer annotations, which means the candidate range for target entities is narrowed from documents to sentences, greatly improving the accuracy of target entity identification. Accurate target entities are undoubtedly crucial for both probe training and the reliability of knowledge reconciliation analysis conclusions.
> > >
> > >
> > > **Regarding the F1 score concerns:**
> > > We understand your concern about whether the F1 score of approximately 50 on HotpotQA indicates that the probe is insufficiently accurate. We would like to clarify that the probe is indeed accurate, and the lower performance is due to the model's insufficient capability in correctly attending to question-relevant parts in the context. Specifically:
> > >
> > > (1) **The performance gap between HotpotQA and SQuAD is due to HotpotQA being more challenging.** As shown in Table 1, HotpotQA is a multi-hop QA dataset where answering questions requires gathering information from two documents scattered across different positions in the prompt. More importantly, the attribution granularity for HotpotQA is at the sentence level, meaning the model needs to accurately locate which specific sentence contains the answer. In contrast, questions in SQuAD are all single-hop problems, and the attribution granularity is at the paragraph level, meaning the model only needs to find the paragraph containing the answer without further specifying the exact sentence.
> > >
> > > **Table 1: Comparison of Dataset Difficulty**
> > > | Comparison Dimension | TriviaQA | SQuAD | HotpotQA |
> > > |---------------------|----------|-------|----------|
> > > | Average Input Length | 4503.0 | 1353.5 | 1366.2 |
> > > | Question Type | Multi-hop | Single-hop | Multi-hop |
> > > | Attribution Granularity | Paragraph-level | Paragraph-level | Sentence-level |
> > > | Attribution F1 (LLaMA3.1) | 62.8 | 91.2 | 51.2 |
> > >
> > > (2) **The probe is accurate**, as evidenced by the following:
> > >
> > > - Strong generalization capability: In our attribution experiments, **the probe was trained only on HotpotQA, and the F1 score of approximately 90 on SQuAD represents zero-shot evaluation results** (we will emphasize this point in the revised paper). This strong cross-dataset generalization indicates that the probe has learned general, essential entity identification patterns rather than dataset-specific surface features.
> > > - Performance differences reflect task difficulty rather than probe deficiencies: The contrast between the low F1 (around 50) on HotpotQA and high F1 (around 90) on SQuAD precisely demonstrates that **once the model correctly attends to the right locations in the document, the probe can sensitively detect entity information in the residual stream**. Therefore, the PRAUC metric obtained by the probe on HotpotQA is reliable, as it accurately reflects the true performance of the model's attention mechanism in complex reasoning tasks.
> > >
> > >
> > > We hope this response resolves your questions regarding our experimental design choices. Please feel free to reach out if you need any further clarification or have additional concerns we can address.
> > >
> > > Thank you for your valuable feedback.
> > >
> > > Best regards,
> > >
> > > The Authors

---

> > > > ### Author Response · Authors · 2025-08-07
> > > > **Official Comment by Authors**
> > > >
> > > > Dear reviewer KnAk,
> > > >
> > > > We are grateful for your feedback and thank you again for the time and effort in reviewing our paper. We believe that our responses above fully address the points raised in the initial review and follow up questions. Since the discussion period is nearing its end, we kindly ask that you let us know whether you have any remaining questions or comments.
> > > >
> > > > Thank you!

---

### Official Review · Reviewer_jqeN · 2025-07-02

**Clarity:** 3
**Significance:** 3
**Originality:** 2
**Rating:** 4
**Confidence:** 4

**Summary:**

This paper presents an entity-aware probing framework for investigating how large language models (LLMs) reconcile conflicting parametric and contextual knowledge, particularly in retrieval-augmented generation (RAG) settings. The authors design a lightweight probing mechanism using LoRA to trace entity flow within transformer layers and evaluate its effectiveness in both attribution and knowledge conflict scenarios. They find that attention and MLP layers process parametric and contextual knowledge via distinct mechanisms, and that aligned multi-source knowledge accumulates in a superpositional manner.

**Questions:**

1. **What is the conceptual connection between the relevance discrimination task and knowledge reconciliation?** Relevance seems like a surface property, whereas reconciliation implies *resolution* of conflicting information. Why is the former a good proxy for the latter?

2. **How does parametric knowledge contribute to relevance judgments?** Given that all relevance predictions seem to depend on context (via `D`), is parametric knowledge playing any functional role in this task?

**Ethical Concerns:**

["NO or VERY MINOR ethics concerns only"]

**Final Justification:**

The authors provide new results to validate the generality of their conclusions

**Limitations:**

yes

**Quality:**

3

**Strengths And Weaknesses:**

### Strength

**Timely and Relevant Topic:** The work addresses an important and relatively underexplored problem—how LLMs reconcile conflicting information from parameters vs. retrieval context.

**Broad and Empirical Analysis:** The paper explores multiple probing dimensions (attention head, MLP, residual stream), across multiple model types and benchmarks, contributing a wealth of experimental insights.

---

### Weakness

**Too Narrow Focus on Relevance Discrimination**: A large portion of the paper is dedicated to building and evaluating the entity relevance probe. However, relevance discrimination alone does not explain *how* reconciliation happens. There is a disconnect between the claim of studying reconciliation mechanisms and the design of experiments, which focus more on *detection* than reconciliation.

**Lack of Grounded Real-World Applications**: The knowledge conflict scenarios are synthetic and overly simplified (e.g., single-entity substitutions). There’s little discussion of how this mechanism translates to more complex real-world retrieval settings with temporal or causal dependencies.

**Weak Link Between Relevance and Parametric Knowledge**: It remains unclear how relevance probing reflects the role of **parametric knowledge**. The entity relevance task seems entirely context-driven (i.e., entities in `D` and their alignment with `q`). The paper does not clarify how parametric knowledge contributes to relevance judgments or whether it is required at all.

---

> ### Author Rebuttal · Authors · 2025-07-31
>
> **Dear Reviewer jqeN**,
>
> We greatly appreciate your thoughtful feedback on our work! We are encouraged by your recognition of `the importance and value of our research topic` as well as `the wealth of experimental insights`. We address your questions and concerns as follows:
>
> > **Q1.** What is the conceptual connection between the relevance discrimination task and knowledge reconciliation? Relevance seems like a surface property, whereas reconciliation implies resolution of conflicting information. Why is the former a good proxy for the latter?
>
> **The dynamic changes in entity relevance (i.e., information intensity) across layers directly reflect the model's mechanism for reconciling conflicting information, rather than being a superficial property unrelated to the reconciliation process.** We illustrate how entity relevance can be used to analyze the model's conflict reconciliation mechanism through a demonstrative example.
>
> Consider the query $q$ = "*Who is the current U.S. President?*" Suppose the model's parametric knowledge is $\mathcal{E}_p$ = "Biden" while the external document $D$ contains contextual knowledge $\mathcal{E}_c$ = "Trump". The model now faces a decision: which of these conflicting pieces of knowledge, $\mathcal{E}_p$ or $\mathcal{E}_c$, should be used to answer query $q$?
>
> - If only one type of knowledge can be detected in the attention layer's output, i.e., Attention("Biden", "Trump") → "Trump", this indicates that the attention layer reads prompt=$\mathcal{D}+q$ but only routes one type of knowledge for answer generation. In this case, conflict reconciliation occurs at the attention layer.
> - If both types of knowledge are routed by the attention layer, but the information intensity of one type rapidly decays in the residual stream at the position where the answer token is generated, i.e., ("Biden", "Trump") → MLP$_i$ → ("Trump"), this suggests that the MLP layer plays the role of conflict reconciler in the residual stream.
>
> (The two hypothetical scenarios above are for illustrative purposes only; the actual conflict reconciliation mechanism is more complex than these two cases. See Section 5.4 of the paper for further discussion.)
>
> > **W1**. Too Narrow Focus on Relevance Discrimination: A large portion of the paper is dedicated to building and evaluating the entity relevance probe. However, relevance discrimination alone does not explain how reconciliation happens. There is a disconnect between the claim of studying reconciliation mechanisms and the design of experiments, which focus more on detection than reconciliation.
>
> In our response to Q1, we clarify how relevance metrics explain reconciliation mechanisms, which we believe addresses your concerns regarding Weakness 1. The experimental design of this paper consistently centers on studying reconciliation mechanisms and provides valuable insights into these mechanisms. This has been acknowledged by Reviewer 6M6N (Strengths 3, 4, 5) and Reviewer oNGe (Strengths 1, 2, 3). Furthermore, probe construction is a necessary means to measure entity relevance (i.e., the information strength of knowledge encoded in model representations), and the probe constructed in this paper differs significantly from classical probes in technical aspects. Therefore, we believe it is necessary to describe the construction method and evaluate its validity. This point has also been recognized by Reviewer KnAk (Strength 3).
>
> >**W2**. Lack of Grounded Real-World Applications: The knowledge conflict scenarios are synthetic and overly simplified (e.g., single-entity substitutions). There’s little discussion of how this mechanism translates to more complex real-world retrieval settings with temporal or causal dependencies.
>
> Thank you for your valuable feedback. We completely agree that the applicability of research work to real-world tasks is an important evaluation criterion. Regarding the concerns you raised, we would like to further clarify the real-world foundations and design considerations of our work:
>
> **(1) LLM-based Knowledge Conflict Generation vs. Simple String Replacement**:
>
> Our knowledge conflict scenario generation is based on LLM chain-of-thought (CoT) reasoning, which is the widely adopted standard method for constructing knowledge conflict scenarios in current research [1,2,3]. This approach is fundamentally different from the simple rule-based string replacement methods used several years ago [4,5,6,7]. When LLMs construct knowledge conflicts, they consider semantic coherence between contexts, different expressions of the same entity within contexts, and logical associations between multi-hop evidence—capabilities that cannot be achieved through simple string replacement.
>
> **(2) Diverse Conflict Types Beyond Simple Entity Substitution**:
>
> Our data construction encompasses not only simple entity replacement conflicts but also complex scenarios including temporal and causal conflicts. Specifically, we construct knowledge conflict scenarios based on HotpotQA, which inherently possesses multi-hop reasoning characteristics. Modifications to entities in the reasoning chain or their related attributes directly impact reasoning outcomes. Here is an example of temporal conflict:
>
> Query: *Which American rock band was formed first, Soil or Drowning Pool?*
>
> Evidence A: *Soil, often typeset as SOiL, is an American rock band that was formed in Chicago, Illinois in 1997.*
>
> Evidence B: *Drowning Pool is an American rock band formed in Dallas, Texas in 1996.*
>
> Answer: *Drowning Pool.*
>
> After analyzing the question and related evidence, the LLM modifies the formation times of both bands as follows:
>
> Revised Evidence A: *Soil, often typeset as SOiL, is an American rock band that was formed in Chicago, Illinois in 1995.*
>
> Revised Evidence B: *Drowning Pool is an American rock band formed in Dallas, Texas in 1998.*
>
> Since the temporal sequence has been reversed, this affects the answer judgment:
>
> Revised Answer: *Soil*
>
> **(3) Trade-off Between Synthetic and Real Data**:
>
> Relying entirely on collecting real-world conflict samples would be prohibitively expensive. Although synthetic data differs from real data, this represents a feasible and effective solution for batch construction of knowledge conflicts at present.
>
>
> [1] Adaptive Chameleon or Stubborn Sloth: Revealing the Behavior of Large Language Models in Knowledge Conflicts.
>
> [2] CONFLICTBANK: A Benchmark for Evaluating Knowledge Conflicts in Large Language Models.
>
> [3] Intuitive or Dependent? Investigating LLMs' Behavior Style to Conflicting Prompts.
>
> [4] Context matters: A pragmatic study of PLMs' negation understanding.
>
> [5] BeliefBank: Adding memory to a pre-trained language model for a systematic notion of belief.
>
> [6] Entity-based knowledge conflicts in question answering.
>
> [7] Context-faithful prompting for large language models.
>
>
> >**Q2&W3**. How does parametric knowledge contribute to relevance judgments? Given that all relevance predictions seem to depend on context (via D), is parametric knowledge playing any functional role in this task?
>
> Since Q2 is a derivative question of W3, we address both concerns together in this response.
>
> The comparison between Figure 4(c) and Figure 5(c) in our paper directly demonstrates the impact of parametric knowledge on relevance judgment. In Figure 4(c), the lines ($\mathcal{D}_{cf}\rightarrow\mathcal{E}\_f$) and ($\mathcal{D}\_{cf}\rightarrow\mathcal{E}\_{cf}$) represent the probing of factual knowledge $\mathcal{E}\_{f}$ and counterfactual knowledge $\mathcal{E}\_{cf}$ from the counterfactual context $\mathcal{D}\_{cf}$, respectively. Although factual knowledge $\mathcal{E}\_{f}$ is never mentioned in the counterfactual context $\mathcal{D}\_{cf}$, we can still observe rapid growth in factual knowledge information intensity in the residual stream near the output layers. The only possible source of this information intensity is the injection of parametric knowledge from the model.
>
> From the perspective of information intensity magnitude, the factual knowledge in Figure 4(c) is weaker than the counterfactual knowledge. This aligns with the LLM's actual response behavior, as shown in Table 1, where 54.6% of model responses follow counterfactual knowledge when the prompt contains only the counterfactual document $\mathcal{D}\_{cf}$. However, when both counterfactual document $\mathcal{D}\_{cf}$ and factual document $\mathcal{D}\_{f}$ are provided in the prompt, the situation reverses. As shown in Figure 5(c), the $\mathcal{E}\_{f}$ from parametric knowledge injection and the contextual knowledge $\mathcal{E}\_{f}$ described in the factual document form a superposition, with information intensity directly exceeding that of $\mathcal{E}\_{cf}$. This also corresponds to the model's actual behavior in Table 1, where 56.5% of model responses follow factual knowledge, far exceeding the 27.7% that follow counterfactual knowledge. Without the information intensity from parametric knowledge injection, the difference between the two would not be so significant.
>
>
> **Table 1: Model response behavior statistics under different document input settings. Data from experiments in Appendix F of the paper.**
> | Context Type | Model Response as Factual Knowledge | Model Response as Counterfactual Knowledge | Other |
> |-|-|-|-|
> | Only counterfactual document $\mathcal{D}\_{cf}$ | 179 (22.7%) | 430 (54.6%) | 179 (22.7%) |
> | Both counterfactual document $\mathcal{D}\_{cf}$ and factual document $\mathcal{D}\_{f}$ provided | 445 (56.5%) | 218 (27.7%) | 125 (15.9%) |
>
> In fact, the training data for the probes also includes predicting the relevance of factual entities based on counterfactual documents—entities that are not mentioned in the documents. Therefore, the training of relevance prediction does not rely entirely on the alignment between entities in $D$ and the query $q$.

---

> > ### Comment · Reviewer_jqeN · 2025-08-06
> >
> > Thanks for the response. I can see that the relevance test can reflect the reconciliation mechanism, but it's likely to reach biased conclusions if it only considers results in this test scenario, as reconciliation happens more than relevance awareness. I appreciate the current efforts to clarify this point, but I think at least another type of evaluation scenario needs to be applied to show that the reached conclusions can be applied to general cases where reconciliation happens.

---

> > > ### Comment · Reviewer_jqeN · 2025-08-06
> > >
> > > I think an experiment about the model's reconciliation of its original generation style and the system prompt's instruction can provide a different viewpoint of the claimed issue, and thus address my concerns.

---

> > > > ### Author Response · Authors · 2025-08-07
> > > > **Response to Reviewer Comments**
> > > >
> > > > **Dear Reviewer jqeN,**
> > > >
> > > > Thank you very much for taking the time to carefully review our paper and for raising these important concerns. Following your suggestion, we examine the model's knowledge reconciliation behavior from a different perspective while maintaining the LLM's original generation mode.
> > > >
> > > > To achieve this, we zero out the top-10 context heads and memory heads identified in our paper. According to the knowledge reconciliation mechanism we discover, this intervention affects the routing of contextual knowledge and parametric knowledge, which in turn influences their accumulation in the residual stream.
> > > >
> > > > The results in Table 1 demonstrate that this practice causally impacts the reconciliation decisions under the LLM's original generation mode: when context heads are removed, the model's reliance on parametric knowledge increases from 22.7% to 34.9%, while its reliance on contextual knowledge decreases from 54.6% to 43.6%. Conversely, when memory heads are removed, the model's reliance on contextual knowledge increases to 65.5%, while parametric knowledge reliance drops significantly to 9.2%. This provides evidence that the reconciliation mechanism we identify operates beyond the specific relevance test scenario and generalizes to original generation mode.
> > > >
> > > > We believe this new perspective helps address your concerns about the generalizability of our findings.
> > > >
> > > > **Table 1:The causal impact of context heads and memory heads on model output behavior**
> > > > | Experimental Setting | Response Consistent with Parametric Knowledge | Response Consistent with Contextual Knowledge | Other |
> > > > |---------------------|-------------------------------------------|--------------------------------------------------|-------|
> > > > | $\mathcal{D}\_{cf}$ | 22.7% | 54.6% | 22.7% |
> > > > | $\mathcal{D}\_{cf}$ w/o Context Head | 34.9% | 43.6% | 21.5% |
> > > > | $\mathcal{D}\_{cf}$ w/o Memory Head | 9.2% | 65.5% | 25.3% |
> > > >
> > > > We use the following prompt format, where parametric (factual) answers, contextual (counterfactual) answers, and other responses (including "I don't know" and "Other") are randomly assigned to the four options to ensure unbiased evaluation.
> > > >
> > > > >\## Document: {Document_Content}
> > > > >
> > > > >You are a helpful QA assistant. Please read the document above and answer the following question. Please reason step by step, and put your final answer within \boxed{}.
> > > > >
> > > > >\## Question: {Question_Content}
> > > > >
> > > > >\## Options:
> > > > >A: {A_Content},
> > > > >B: {B_Content},
> > > > >C: {C_Content},
> > > > >D: {D_Content}
> > > >
> > > >
> > > > We hope this additional experiment from a different angle addresses your concerns. Please let us know if you have any other questions or concerns that we can address.

---

> > > > > ### Comment · Reviewer_jqeN · 2025-08-07
> > > > >
> > > > > Thanks for the explanation and the new results! I think this result supports the generality of the discoveries in this paper. (But more types of generation styles can still be considered to be involved in experiments) In response to the quality improvement in this paper. I have **raised my score** for a reflection.

---

> > > > > > ### Author Response · Authors · 2025-08-07
> > > > > > **Official Comment by Authors**
> > > > > >
> > > > > > It's encouraging to know that our additional analysis has effectively addressed your previous concerns. We sincerely value your continued guidance and will do our utmost to strengthen the quality of our submission.

---

### Decision · Program_Chairs · 2025-09-17

**Decision:**

Accept (spotlight)

**Comment:**

This paper presents a comprehensive study on how LLMs reconcile internal, parametric knowledge with external, contextual knowledge provided in RAG settings. The problem of knowledge reconciliation and attribution is of critical importance for the reliability and trustworthiness of modern AI systems. To investigate this, the authors introduce a novel and clever entity-aware probing framework that uses special tokens and a lightweight LoRA adapter to trace the flow of information related to specific entities through the transformer layers.

The reviewers raised valid concerns regarding the experimental design, the potential influence of the probing methodology, and the generalizability of the conclusions. However, the authors provided an exemplary rebuttal, including additional causal experiments where they ablated specific attention heads to observe the direct impact on model outputs. This new evidence substantially strengthened the paper's claims and successfully addressed the reviewers' concerns, leading to a consensus of acceptance.

This is a well-executed and insightful paper that makes a significant contribution to our understanding of knowledge processing in LLMs. The work is timely, the methodology is sound, and the conclusions are well-supported. While the work primarily uses synthetically generated conflicts, this approach provides a controlled environment necessary for this type of detailed analysis. The findings lay a strong foundation for future investigations into more complex, real-world knowledge conflicts. I encourage the authors to continue their work in this promising direction.

I recommend accepting this paper.